# WARD: PROVABLE RAG DATASET INFERENCE VIA LLM WATERMARKS

**Nikola Jovanović, Robin Staab, Maximilian Baader, Martin Vechev**
ETH Zurich
{nikola.jovanovic, robin.staab, mbaader, martin.vechev}@inf.ethz.ch

## ABSTRACT

RAG enables LLMs to easily incorporate external data, raising concerns for data owners regarding unauthorized usage of their content. The challenge of detecting such unauthorized usage remains underexplored, with datasets and methods from adjacent fields being ill-suited for its study. We take several steps to bridge this gap. First, we formalize this problem as (black-box) RAG Dataset Inference (*RAG-DI*). We then introduce a novel dataset designed for realistic benchmarking of RAG-DI methods, alongside a set of baselines. Finally, we propose WARD, a method for RAG-DI based on LLM watermarks that equips data owners with rigorous statistical guarantees regarding their dataset's misuse in RAG corpora. WARD consistently outperforms all baselines, achieving higher accuracy, superior query efficiency and robustness. Our work provides a foundation for future studies of RAG-DI and highlights LLM watermarks as a promising approach to this problem.

## 1 INTRODUCTION

Retrieval-augmented generation (RAG) has emerged as a popular approach to mitigate limitations of large language models (LLMs) such as hallucinations, the high cost of adapting to new knowledge via fine-tuning, and the inability to back up claims by sources (Lewis et al., 2020). By integrating retrieval, LLMs gain in-context access to large corpora of high-quality, up-to-date data, enabling them to generate more accurate and source-supported responses. To maintain relevance, RAG providers must continuously update their corpus with new data. However, this raises concerns regarding the unauthorized usage of documents, particularly when publicly available documents are used without the owner's permission (Grynbaum & Mac, 2023; Wei et al., 2024a); see App. A for a more elaborate discussion of this issue and its prevalence in practice. Crucially, there is currently no way to conclusively prove such unauthorized usage by a RAG system, and enforce an opt-out by the owner.

**RAG Dataset Inference (RAG-DI)** We formalize this problem as *RAG Dataset Inference (RAG-DI)*, where a data owner aims to detect unauthorized inclusion of their dataset in a RAG corpus via black-box queries (Fig. 1). In the first comprehensive study of this problem, we observe that existing datasets, used in adjacent works on RAG privacy, are not suitable for RAG-DI. First, the samples in these datasets may have been used in contemporary LLM training, complicating realistic evaluations where RAG corpora consist of new data. Second, these datasets do not model *fact redundancy*, a key property of real-world RAG, where multiple documents have similar content, either due to scraping data from various sources, e.g., news (Gao et al., 2023), or due to chunking. Another challenge in studying RAG-DI stems from the lack of baselines applicable in a realistic black-box setting.

**Foundations for RAG-DI** In this work, we take multiple steps to bridge these gaps: First, we introduce FARAD, a new dataset specifically designed for RAG-DI evaluation under realistic conditions. FARAD contains fictional articles that are by design not part of any LLM training data, and can enable evaluations under fact redundancy, enabling accurate assessment of RAG-DI methods. Second, we adapt prior work on RAG Membership Inference Attacks (MIAs) (Li et al., 2024; Anderson et al., 2024) to the RAG-DI problem, and propose a simple baseline FACTS. In our evaluation on FARAD, we find that (i) despite its extreme simplicity, FACTS outperforms other baselines in settings with no fact redundancy, further underscoring the drawbacks of existing datasets, and (ii) when fact redundancy is present, no baseline achieves satisfactory performance. This highlights the need for novel approaches capable of reliably identifying unauthorized usage of documents in RAG corpora.

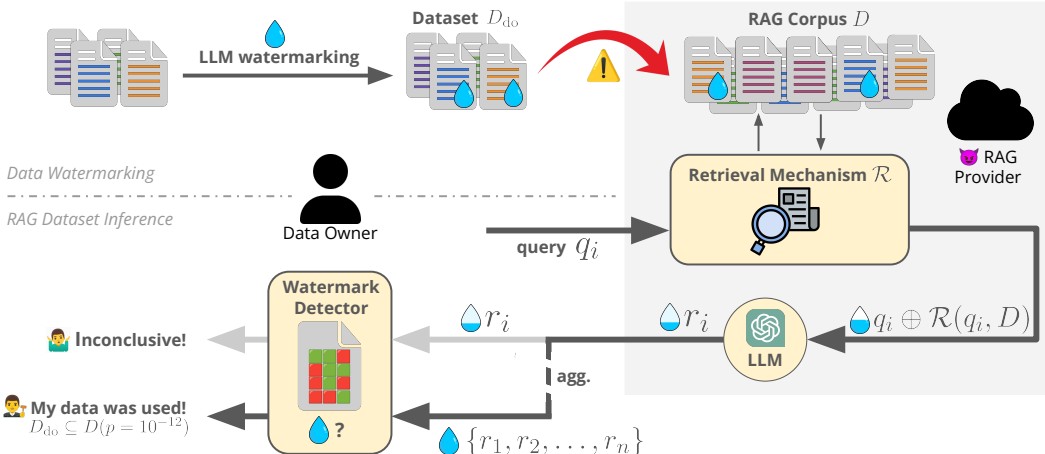

Figure 1: Overview of RAG Dataset Inference using WARD, our method based on LLM watermarks.

**LLM Watermarks as a reliable RAG-DI method**    To this end, we introduce WARD, a RAG-DI method that protects the data owner's dataset by imprinting LLM watermarks (Kirchenbauer et al., 2023; Kuditipudi et al., 2024). As Fig. 1 illustrates, given a limited number of black-box queries $q_i$ to the retrieval-augmented LLM, the data owner can detect even small traces of the watermark across responses $r_i$, and for the first time obtain rigorous statistical guarantees regarding the usage of their dataset in the RAG corpus, enabling them to effectively audit the RAG provider. In our experiments on FARAD, we show that WARD consistently outperforms all baselines across a variety of challenging settings, showing high rate of true positives (often 100%), with no false accusations. Further, due to its robustness across settings, and the fact that it engages only in natural-looking interactions with the retrieval-augmented LLM, WARD retains its performance even under attempts by the RAG provider to prevent unintended uses of the system. These results underscore the effectiveness of WARD in enforcing data usage policies and protecting data owners' rights in the context of RAG systems.

**Main contributions**    We make the following key contributions:

- We formalize a previously unexplored problem, RAG Dataset Inference (RAG-DI), where a data owner aims to detect unauthorized usage of their dataset in a RAG system (§3).
- We facilitate research on this problem by (i) proposing a new dataset FARAD, specifically designed for benchmarking RAG-DI methods under realistic conditions, and (ii) introducing an initial set of baseline methods (§3.1 and §3.2).
- We propose LLM watermarks as a way to provably, robustly and reliably detect unauthorized data usage in RAG corpora, introducing WARD as a novel RAG-DI method (§4).
- Our experimental evaluation in a wide range of settings affirms the fundamental limitations of existing datasets and all RAG-DI baselines, and demonstrates the effectiveness of WARD, which consistently shows high accuracy, query efficiency, and robustness (§5). Our source code and the FARAD dataset are publicly available at https://github.com/eth-sri/ward.

## 2    BACKGROUND

**Retrieval-augmented generation (RAG)**    RAG is a common way to enhance LLMs: For a given user query $q$, the $k \in \mathbb{N}$ most relevant documents $D_q = \mathcal{R}(q, D) \subseteq D$ are retrieved from a corpus $D$, using a retrieval method $\mathcal{R}$ (Gao et al., 2023). The query $q$ is generally combined with $D_q$, and fed into an LLM to generate a more factual response $r = \mathcal{M}(q, D_q)$. Expanding $D$ also enables access to new information without costly retraining (Lewis et al., 2020). RAG is especially suitable for domains where new information is generated often, e.g., news articles or software documentation. In this work we use $\mathcal{M}^{\star}$ to denote a *RAG system*, i.e., a retrieval-augmented LLM $\mathcal{M}$.

**LLM watermarking**    LLM watermarks enable model owners to provably and reliably track text generated by their LLM. In this work, we focus on the prominent *red-green watermarks* (Kirchenbauer

et al., 2023). During text generation, at each step $t$, the token vocabulary $V$ is split into two parts, $\gamma|V|$ *green* (encouraged) tokens, and $(1-\gamma)|V|$ *red* (discouraged) tokens, for some $\gamma \in [0, 1]$. Given *context width* $h$, the split is commonly a function of tokens at positions $t - h, \dots, t$, as well as a secret salt. To add the watermark, the logits of green tokens are increased by $\delta \in \mathbb{R}_{\geq 0}$, boosting their probability of being sampled. The watermark is detected using a statistical test, based on the expectation that non-watermarked text of length $T$ has $\gamma T$ green tokens. Namely, we use the z-score

$$z = \frac{|s|_g - \gamma T}{\sqrt{\gamma(1-\gamma)T}}, \tag{1}$$

where $|s|_g$ is the number of green tokens in a given text $s$, and $T = |s|$. From here, we derive the p-value $p = 1 - \Phi(z)$, where $\Phi$ is the CDF of the standard normal distribution, and consider the text watermarked if $p < \alpha$ for some threshold $\alpha$. While not perfectly robust, these watermarks generally persist under moderate text transformations, such as paraphrasing or segment omission (Piet et al., 2023; Kirchenbauer et al., 2024; Sander et al., 2024), making them suitable for our setting, where our goal will be to propagate the watermark signal through the RAG pipeline, as we will describe in §4.

**RAG membership inference attacks**  While RAG is a relatively novel concept, recent work already studies membership inference attacks (MIAs) in this setting, proposing two methods that we denote SIB (Li et al., 2024) and AAG (Anderson et al., 2024). A MIA's goal is to output $\text{mi}(d, \mathcal{M}^\star) = 1$ if a document $d$ is part of the retrieval corpus $D$ of a RAG system $\mathcal{M}^\star$, or $\text{mi}(d, \mathcal{M}^\star) = 0$ otherwise, based only on queries to $\mathcal{M}^\star$. To this end, SIB queries $\mathcal{M}^\star$ with $q$, a prefix of $d$, to obtain the response $r = \mathcal{M}(q, D_q)$. Then, it computes two scores: the cosine similarity between the embeddings of $d$ and $r$, and the perplexity of $r$, outputting $\text{mi}(d, \mathcal{M}^\star) = 1$ if the similarity is above a threshold $\theta_{\text{similarity}}$, and the perplexity is below a threshold $\theta_{\text{perplexity}}$, both trainable parameters. Note that this requires gray-box access to $\mathcal{M}^\star$ for perplexity computation. The other method, AAG, directly prompts $\mathcal{M}^\star$ to answer if $d$ is in the context. If it replies positively, they set $\text{mi}(d, \mathcal{M}) = 1$. In §3, we will introduce the black-box RAG dataset inference setting, and adapt both baselines to it. Notably, while dataset inference was studied alongside MIA for training data (see §6), no prior work studies it for RAG.

## 3 RAG DATASET INFERENCE

We formalize the problem of *RAG Dataset Inference* (*RAG-DI*), and present our contributions aimed at facilitating studies of this problem. In §3.1, we make the case that existing datasets commonly used for adjacent tasks (e.g., RAG MIA) are fundamentally unsuitable for RAG-DI, and propose a new dataset in an attempt to address these shortcomings. In §3.2, we establish a set of baselines for RAG-DI, by adapting RAG MIA work introduced in §2, and proposing a simple baseline, FACTS.

**The RAG-DI problem**  The key entities in RAG-DI are the *data owner*, who aims to protect their $n$-document dataset $D_{\text{do}}$ from unauthorized usage in a RAG corpus, and the *RAG provider*, who exposes black-box access to their retrieval-augmented LLM $\mathcal{M}^\star$, which uses a corpus $D$. The data owner's goal is to determine if $D_{\text{do}} \subseteq D$, i.e., whether their data was secretly included in the corpus. To this end, they may proactively modify $D_{\text{do}}$ before publishing it, and can query $\mathcal{M}^\star$ in a black-box way, aiming to minimize the number of such queries. Crucially, the data owner makes a single dataset-level decision, as opposed to document-level decisions of MIAs. Formally, a RAG-DI method $di$ should output $\text{di}(D_{\text{do}}, \mathcal{M}^\star) = 1$ if $D_{\text{do}} \subseteq D$ (*IN* case) and 0 otherwise (*OUT* case).

### 3.1 A DATASET SUITABLE FOR RAG-DI

To enable suitable evaluation of RAG-DI methods, we require a dataset of documents with the following properties. First, we aim to match a key use-case of RAG (as described in §2) where up-to-date knowledge is added to $D$ instead of costly repeated fine-tuning of $\mathcal{M}$. To model this, our documents should provably not be part of the training data of $\mathcal{M}$, i.e., of current open/closed LLMs, as those will be used to instantiate $\mathcal{M}$ when studying RAG-DI. Second, to model the practical case where knowledge is redundant and spread across multiple sources (e.g., news articles, a common motivating example for RAG (Gao et al., 2023)), the dataset should contain documents with overlapping topics and information (*fact redundancy*). As we will empirically demonstrate in §5, while fact redundancy is more realistic, it makes RAG-DI significantly harder.

**Current state** As there is no prior work on RAG-DI, we turn to related work on RAG privacy (Anderson et al., 2024; Li et al., 2024; Zeng et al., 2024), including MIAs introduced in §2. We observe that evaluations in these works rely primarily on *EnronEmails* (Klimt & Yang, 2004) and *HealthcareMagic* datasets (Zeng et al., 2020; Mrini et al., 2021), motivated by the presence of PII in their samples. However, it can not be ruled out that these datasets were used to train contemporary LLMs, as e.g., *EnronEmails* has been publicly available since 2004. More importantly, fact redundancy is by design not satisfied in either of these cases. These shortcomings motivate us to construct a new dataset, FARAD (*Fact-Redundant Article Dataset*), tailored to RAG-DI.

**The FARAD dataset** FARAD consists of a number of *groups*. Each group contains articles that share a topic and a significant amount of information, but are independently written by a different (LLM) author. As our data source we use *RepLiQA* (Monteiro et al., 2024), that contains articles about fictional entities and events, which by design ensures that this knowledge was not present in any LLM training data. *RepLiQA* is released gradually—to create FARAD, we use split 0 as the only one available at the time of writing, but plan to expand this to future splits.

Each *RepLiQA* article is a *source* for one of our groups—we use the pipeline illustrated in Fig. 2. First, we prompt GPT4O to distill the information content of the article into 5 self-contained *key facts*, crucial to understanding the article, and 10 self-contained *additional facts*, that are present in the article but not essential to its key message (see App. F.2). Next, to create a group, we sample an author model from the set $\mathcal{A} = \{$GPT4O, CLAUDE3.5-SONNET, LLAMA3.1-

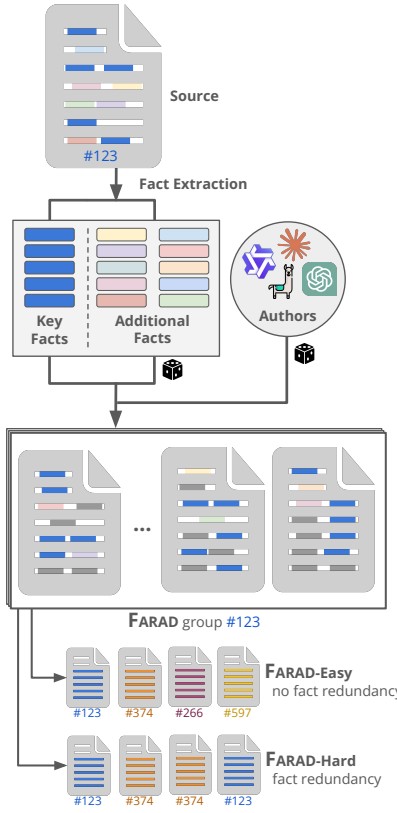

Figure 2: Overview of the generation pipeline of FARAD, and the resulting *Easy* and *Hard* evaluation settings.

405B, QWEN1.5-110B$\}$ of state-of-the-art LLMs. The selected model writes a 500–1000 word article that must include all 5 key facts, 2 randomly sampled additional facts, and is encouraged to invent additional quotes, hypotheses, or personal opinions, as long as they do not contradict any present fact. The diversity of authors, sampled facts, and (desirable) hallucinated content results in a varied set of fictional articles grounded in the same core knowledge, which satisfies our earlier requirements. While this pipeline can be extended to larger/more groups, more authors, and different fact combinations, we limit FARAD to 3591 groups (3391 for testing, and 200 for training) of 4 articles each, one per author from $\mathcal{A}$.

**Use in RAG-DI evaluation** In our experiments in §5, we use FARAD to create two evaluation settings: *Easy* and *Hard*, illustrated in Fig. 2 (bottom). The *Easy* setting follows the traditional setup described above where facts are uniquely represented, by always using at most 1 article per FARAD group. The *Hard* setting introduces fact redundancy by always including all 4 articles from a group. As we demonstrate shortly, the *Easy* setting is solvable by a simple baseline, while the *Hard* setting, as the realistic one and our main focus, is a significant challenge for all baseline methods. In App. E.1 we discuss potential improvements to our data generation pipeline as items for future work.

## 3.2 RAG-DI BASELINES

We proceed to establish an initial set of baselines for the RAG-DI problem. For this we adapt existing RAG MIAs, AAG and SIB, introduced in §2, and additionally propose a simple baseline, FACTS.

**Adapting RAG MIAs** By design, the existing RAG MIAs make document-level decisions, i.e., they decide if a *single document* is in the RAG corpus ($\text{mi}(d, \mathcal{M}^\star) = 1$) or not ($\text{mi}(d, \mathcal{M}^\star) = 0$). We extend this to dataset-level decisions of RAG-DI in an (empirically) optimal way. First, we apply a given method to each $d \in D_{\text{do}}$ to obtain $n$ binary decisions, and set $s(D_{\text{do}}, \mathcal{M}^\star) = \frac{1}{n} \sum_{d \in D_{\text{do}}} \text{mi}(d, \mathcal{M}^\star)$.

We then instantiate several *IN* and *OUT* cases of RAG-DI using the training set, and compute the corresponding $s_{in}$ and $s_{out}$. We finally make a dataset-level statement $\mathrm{di}(D_{do}, \mathcal{M}^\star) = 1$ if $s(D_{do}, \mathcal{M}^\star) > \frac{1}{2}(s_{in} + s_{out})$ and $\mathrm{di}(D_{do}, \mathcal{M}^\star) = 0$ otherwise. During this process, for SIB, we also grid search over its trainable parameters $\theta_{similarity}$ and $\theta_{perplexity}$ (as in Li et al. (2024)), choosing values that maximize $s_{in} - s_{out}$. We note that SIB is proposed as a gray-box method; we adapt it to our black-box setting by using an auxiliary language model to estimate the perplexity of responses.

**The FACTS baseline** To substantiate our point about the effect of fact multiplicity on the hardness of RAG-DI, we introduce a simple baseline, FACTS. As the above RAG MIAs, FACTS is a document-level method. Given a document $d$, FACTS prompts an auxiliary LLM to generate a single question that is only answerable by reading $d$. Then, $\mathcal{M}^\star$ is prompted with that question, and if it deems it unanswerable, we set $\mathrm{mi}(d, \mathcal{M}^\star) = 0$, and set it to 1 otherwise. We aggregate such document-level decisions into a corpus-level decision by fitting a threshold on the training set, as described above. In §5, we will demonstrate that this simple method is sufficient to solve the *Easy* setting more reliably than other baselines, yet is, due to fact multiplicity, extremely unreliable in the *Hard* setting.

## 4 LLM Watermarking as an Effective RAG-DI Method

Before describing our proposed RAG-DI method based on LLM watermarks, we first outline three key design requirements that a desirable RAG-DI method should fulfill. We require the following:

1. *Monotonicity*. With more queries to the retrieval-augmented LLM $\mathcal{M}^\star$, the accuracy of the method's predictions should consistently improve, preferably at a high rate.

2. *Guarantees*. The method should be able to provide a statistical guarantee for its decision, with exceedingly rare and well-controlled Type 1 errors, as falsely accusing RAG providers is highly undesirable in practice, and undermines the trust in the method.

3. *Robustness*. The method should maintain high accuracy under diverse evaluation settings, including attempts by the RAG provider to actively conceal unauthorized data usage.

In App. A we motivate these requirements in detail by reflecting on the context around RAG-DI. As we demonstrate in §5, all RAG-DI baselines introduced in §3.2 violate *all* requirements to some extent. To address this, we propose WARD (*Watermarking for RAG-DI*), a proactive RAG-DI method that is based on LLM watermarks, and discuss why it is likely to fulfill all stated desiderata.

**WARD: RAG-DI via LLM watermarking** We assume the data owner has protected each $d_i \in D_{do}$ by embedding an LLM watermark either via a human-in-the-loop procedure or (as in this work) by rephrasing each document with a watermarked LM. While, in principle, any LLM watermark can be applied, we focus on popular red-green watermarks (see §2). In §5.3, we confirm that this results in quality texts, faithful to the original ones. To audit the RAG provider's corpus, for each $d_i \in D_{do}$, WARD generates an open-ended content-related question $q_i$, and queries $\mathcal{M}^\star$. If $d_i \in D$ (*IN* case), we expect the retrieval method $\mathcal{R}$ to introduce watermarked content from $d_i$ into the LLM context. As noted in §2 and validated in §5, the robustness of watermarks to text transformations is then sufficient to propagate the traces of the signal to $r_i = \mathcal{M}^\star(q_i)$, the final response of the LLM.

**Boosting a weak signal** Requiring that each $r_i$ is flagged as watermarked, i.e., has watermark detector p-value $p < \alpha$, would be a strong assumption, as the watermark signal is likely to degrade throughout the RAG pipeline. However, this is not necessary for WARD to be effective. Instead, following Sander et al. (2024), after $n$ queries we compute a *joint* p-value $R = \{r_1, \ldots, r_n\}$, directly corresponding to the null hypothesis *"the data owner's dataset $D_{do}$ is not in the RAG corpus $D$"*.

This joint p-value can satisfy $p < \alpha$, i.e., reject the null hypothesis, even when individual $r_i$ carry only weak watermark signal, that would individually not reject it. To illustrate this, given a desired p-value threshold $\alpha$, Eq. (1) implies that the required ratio of green tokens in $R$ is at least

$$\gamma' \geq \Phi^{-1}(1 - \alpha) \cdot \sqrt{\gamma(1 - \gamma)/|R|^{\oplus}} + \gamma, \tag{2}$$

where $\Phi$ is the standard normal distribution CDF, and $|R|^{\oplus}$ the total length of responses in $R$. This lower bound decreases quickly as $|R|^{\oplus}$ increases: In Fig. 3 we plot the lower bound as a function of $n$, assuming $\forall i\colon |r_i| = 400$ tokens, $\alpha \approx 3 \cdot 10^{-5}$ (z-score of at least 4), and $\gamma = 0.25$, as in our experiments. For $n \geq 100$, if propagation of the watermark through the RAG pipeline increases the ratio of green tokens by only $1\%$, it is already detectable with high confidence. This makes WARD viable, satisfying the *Guarantees* requirement, unlike any baseline. It also contributes to *Monotonicity*: assuming each $d_i$ has a green token ratio of $\gamma''$, which propagated through $\mathcal{M}^{\star}$ reduces to $\gamma' \in (\gamma, \gamma'')$, the p-value for *IN* cases strictly decreases for more queries. In §5.2 we experimentally confirm that WARD scales consistently and efficiently in the number of queries.

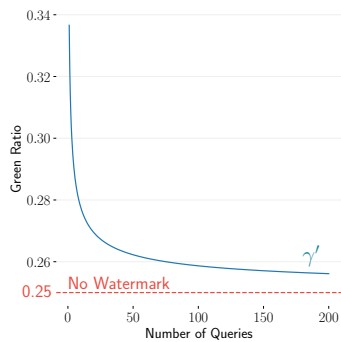

Figure 3: Green ratio to pass watermark detection with given #queries.

**Test validity and signal leakage**  Another requirement for *Monotonicity* is that the p-value for *OUT* cases is well-calibrated, ensuring exceedingly rare false positives. As pointed out in prior work (Kirchenbauer et al., 2023; Sander et al., 2024), to ensure independence between tokens scored by the watermark detector, it is necessary to ignore duplicate $h$-grams across $R$. While this reduces the number of useful tokens per query, our results in §5 show that it does not affect WARD performance.

For our statistical test to be valid, we must ensure that the watermark signal can only originate from the use of $D_{\text{do}}$ in $D$, i.e., that it is not otherwise *leaked* to the responses $r_i$. In particular, we observed that watermarked queries, or queries based on watermarked versions of $d_i$, drastically increase false positives. To confirm that WARD does not suffer from this issue, in §5.2, we verify that the p-values in *OUT* cases are distributed in $[0, 1]$ roughly as expected with no noticeable decrease in $|R|$.

**Scheme choice**  The key parameter of red-green schemes is $h$, the context width. In common applications of LLM watermarks, a low $h$ is not recommended, as it makes the watermark easy to steal via repeated queries (Kirchenbauer et al., 2024; Jovanović et al., 2024). In the context of RAG-DI, this is less of a concern, as the data owner never exposes unconstrained query access to watermarked content, as is the case when watermarking LLMs. Thus, as low $h$ benefits watermark propagation through RAG, we use $h = 2$. This contributes to high *Robustness* of WARD, empirically validated in §5.2. We discuss other scheme parameters and present ablation studies over each in §5.4.

**Practicality**  Another aspect making WARD robust is that it is inconspicuous, as it operates on natural-looking documents, queries, and responses (confirmed in §5.3), through legitimate use of the RAG system—this makes potential attempts by the RAG provider to thwart malicious interactions less effective. This differs from baselines such as AAG, which directly reveal the intention to leak the information about $D$. As a final advantage, we note that, in contrast to all baselines, WARD does not require any training or adaptation to RAG-DI, as LLM watermarks naturally apply to this task.

## 5  EXPERIMENTAL EVALUATION

We evaluate the RAG-DI baselines (§3.2) and WARD (§4) on the FARAD dataset (§3.1). §5.1 presents our main experiment. In §5.2 we focus on desiderata from §4, showing that only WARD does not violate them. In §5.3 we validate several key assumptions, and in §5.4 present additional ablations.

**Setup**  Our experimental setup follows §3: we use FARAD to define two evaluation settings, and in both evaluate *IN* and *OUT* cases, i.e., where the data owner's data *is* (resp. *is not*) contained in $D$. We use $|D_{\text{do}}| = 200$, and $|D| = 800$ for FARAD-*Easy*, and $|D| = 3000$ for FARAD-*Hard* (sampling detailed in App. B.1). We note that WARD only depends on $|D_{\text{do}}|$, but not $|D|$. We use several LLMs as $\mathcal{M}$: GPT3.5, CLAUDE3-HAIKU, and LLAMA3.1-70B, and vary the system prompt: we use a short *naive* prompt (*Naive-P*) with basic RAG instructions, and a longer *defense* (*Def-P*) prompt, which models a RAG provider that instructs the model to not regurgitate sources verbatim, and refuse attempts to learn about the exact LLM context (see App. F.1). Each experiment is run with 5 random seeds. To ensure a controlled setting, in our main experiments, we assume a perfect retrieval system that always retrieves the most relevant documents (see App. B.2 for a detailed explanation). In §5.3,

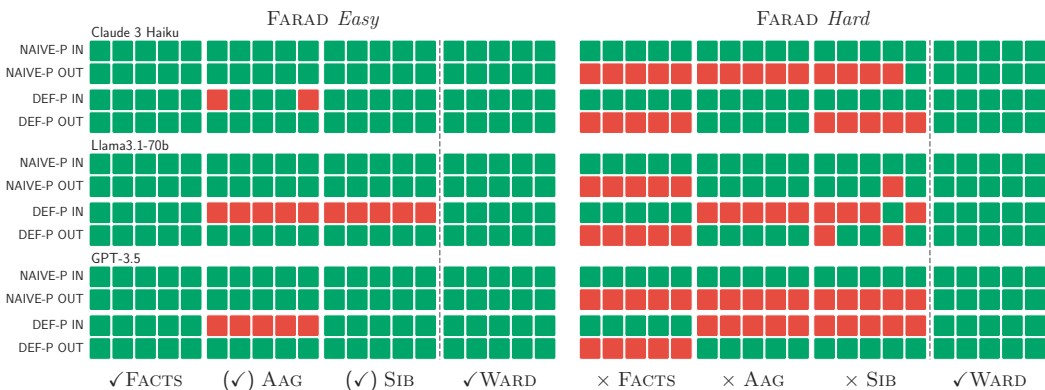

Figure 4: Evaluation of all methods on FARAD in both *Easy* and *Hard* settings, and with both *Naive-P* and *Def-P* system prompts. We run each method with 5 random seeds, resulting in 5 squares. A red square indicates a false negative in the *IN* case, and a false positive in the *OUT* case. All methods perform well in the *Easy* setting, while only WARD consistently performs well in the *Hard* setting.

we show that WARD works equally well on a practical end-to-end RAG. If not specified otherwise (see §5.4), the RAG uses $k = 3$ shots. For methods that utilize auxiliary LMs (including WARD, which starts by watermarking $D_{do}$) we use LLAMA3.1-8B, and to compute the cosine similarity in SIB we use paragraph-level ALL-MINILM-L6-V2 with BERTScore aggregation (Zhang et al., 2020). For WARD, we use *PositionPRF* (Kirchenbauer et al., 2024), $h = 2$, and $\delta = 3.5$, ablating these in §5.4. We list all prompts, and examples of FARAD samples and watermarked documents, in App. F.

## 5.1 MAIN RESULTS

We present our main results in Fig. 4, where we evaluate all RAG-DI baselines and WARD across several settings, models, prompts, and random seeds. We make several key observations.

First, all baselines perform somewhat well in the *Easy* setting (no fact redundancy). However, our extremely simple FACTS baseline obtains perfect results, outperforming both AAG and SIB. This emphasizes that traditional non-redundant datasets (see §3.1) fail to capture the complexity of RAG-DI, and can provide an incomplete view of the capabilities of RAG-DI methods. We also observe that already in the *Easy* setting, straightforward system prompt defenses significantly impact both AAG and SIB, leading to a noticeable increase in false negatives. We further investigate defenses in §5.2.

In the *Hard* setting, all baselines fail to perform consistently, inducing both false positives and negatives. This can be attributed to their shortcomings in handling fact redundancy: FACTS directly relies on facts, while SIB and AAG rely on semantic similarity influenced by factual content. Notably, only WARD achieves 100% accuracy across all settings, models, and system prompts, showing that despite the retrieval of documents with partially overlapping facts, watermarking provides a reliable signal for dataset inference. This backs up our claims regarding the importance of fact-redundancy for realistic RAG-DI evaluation and highlights the potential of watermarking as an approach to RAG-DI.

## 5.2 DESIDERATA

We next demonstrate how baselines violate the desiderata from §4, which is not the case for WARD.

**Monotonicity** As stated in §4, RAG-DI methods should steadily improve with more queries, i.e., $|D_{do}|$ for WARD (see §5.4 for a generalization). We evaluate this by setting $|D_{do}| \in \{20, 40, \ldots, 200\}$ in FARAD-*Hard*, presenting the results of WARD and SIB in Fig. 5. WARD improves consistently with $|D_{do}|$, reaching perfect accuracy across all settings for at most 80 documents. In our extended discussion of efficiency in App. D we show that this translates to inexpensive API costs of below $1, for all closed-source LLMs we consider. In contrast, SIB, besides never reaching full accuracy, exhibits strongly varying accuracy over $|D_{do}|$, often decreasing despite using more queries. This is a consequence of the need to adapt the MIA baselines to the RAG-DI setting—we find similar behavior across all baselines, and present an additional study of their decision thresholds in App. C.6.

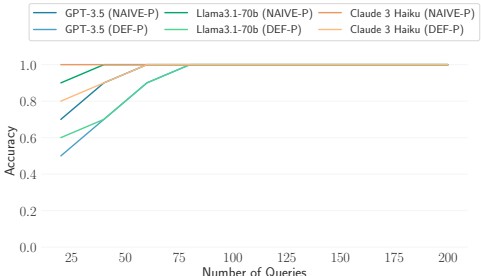 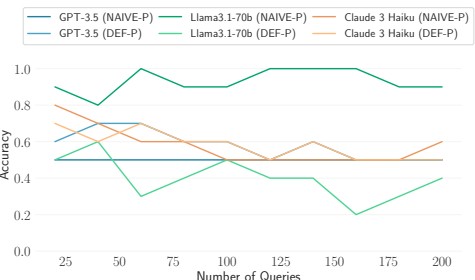

Figure 5: WARD (left) and SIB (right) accuracy as a function of $|D_{\text{do}}|$ (i.e., the number of queries) in the *Hard* setting. WARD consistently improves, while SIB suffers from high variance in accuracy.

Table 1: The p-values of WARD. We report the max p-values for *IN* and min p-values for *OUT* cases.

| Data | Agg. | Claude 3 Haiku | | | | GPT-3.5 | | | | Llama3.1-70b | | | |
| | | DEF-P | | NAIVE-P | | DEF-P | | NAIVE-P | | DEF-P | | NAIVE-P | |
| | | *OUT* | *IN* | *OUT* | *IN* | *OUT* | *IN* | *OUT* | *IN* | *OUT* | *IN* | *OUT* | *IN* |
| FARAD *Easy* | min | $1.90e^{-01}$ | $2.07e^{-267}$ | $1.38e^{-03}$ | $3.47e^{-301}$ | $3.28e^{-01}$ | $9.53e^{-33}$ | $2.13e^{-01}$ | $1.78e^{-78}$ | $7.54e^{-03}$ | $6.20e^{-179}$ | $1.32e^{-01}$ | 0 |
| | max | $6.83e^{-01}$ | $1.57e^{-186}$ | $2.83e^{-01}$ | $2.19e^{-253}$ | $7.72e^{-01}$ | $1.54e^{-27}$ | $9.17e^{-01}$ | $6.56e^{-60}$ | $9.55e^{-01}$ | $1.52e^{-138}$ | $5.05e^{-01}$ | 0 |
| FARAD *Hard* | min | $6.03e^{-02}$ | $1.35e^{-48}$ | $3.82e^{-01}$ | $5.08e^{-74}$ | $2.04e^{-03}$ | $1.12e^{-12}$ | $2.69e^{-01}$ | $1.53e^{-22}$ | $1.18e^{-02}$ | $3.94e^{-24}$ | $6.12e^{-02}$ | $1.22e^{-122}$ |
| | max | $3.99e^{-01}$ | $2.15e^{-25}$ | $7.34e^{-01}$ | $3.16e^{-51}$ | $7.80e^{-01}$ | $8.44e^{-10}$ | $7.92e^{-01}$ | $1.31e^{-15}$ | $7.47e^{-01}$ | $1.10e^{-11}$ | $3.70e^{-01}$ | $2.42e^{-96}$ |

**Guarantees** In contrast to *all* baselines, WARD inherits the guarantees of the watermarking scheme, directly providing a p-value for each decision. In Fig. 6, we see that p-values rapidly decrease with more queries for *IN*, but stay consistently close to the expected value for *OUT*. Further, in Table 1, we show the *max* p-values for *IN* and min p-values for *OUT* in our main experiment, i.e., closest to a false negative/positive. All p-values are orders of magnitude from our decision boundary of $\approx 3e^{-5}$ (z-score of 4), highlighting robustness to Type 1 errors and the fact that LLM watermark signals persist through a RAG pipeline.

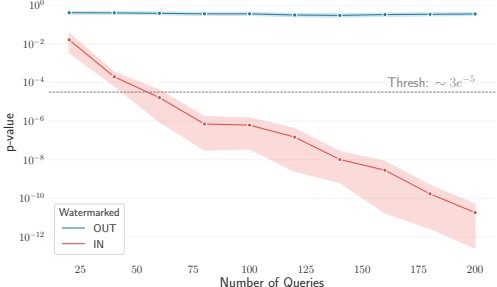

Figure 6: P-values as a function of #queries in *IN* (red) and *OUT* (blue) cases.

**Robustness** Finally, we evaluate the methods' robustness to different settings. In particular, we further examine the *Def-P* system prompt setting, shown in Fig. 4 to often prevent all baselines from obtaining any useful estimates. Notably, both AAG and SIB rely on the model's *willingness* to leak information about its context, either explicitly or by regurgitating similar content. To validate the effect of *Def-P*, we show the longest token-string overlap between the response of $\mathcal{M}^\star$ and the SIB target document (Fig. 7), observing strictly less overlap with *Def-P*, inducing false negatives. Importantly, this does not prevent watermarks from propagating through the RAG pipeline, resulting in a reliable signal for WARD.

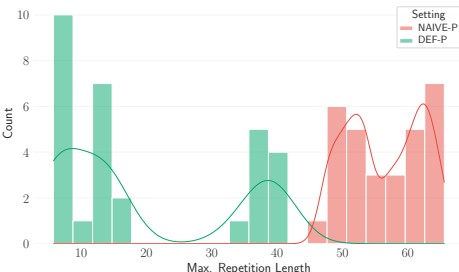

Figure 7: The effect of the *Def-P* sysprompt.

We additionally investigate a defense based on MEMFREE decoding (Ippolito et al., 2023). Namely, we adapt the decoding of $\mathcal{M}^\star$ to strictly prevent $n$-gram overlap with the retrieved documents (using $n = 10$ as in Ippolito et al. (2023)), modeling an even stronger attempt by the provider to protect the RAG corpus. Our results (App. C.5), show that even in this setting, WARD achieves full accuracy. We further discuss the potential for more elaborate defenses against WARD in App. E.2.

## 5.3 ADDITIONAL CONSIDERATIONS

**Modeling retrieval** So far, our experiments have assumed an idealized case of perfect retrieval. We now justify this choice by running WARD on an end-to-end RAG system which uses OpenAI's `text-embedding-3-large` document embeddings, with $k = 3$ and cosine similarity metric.

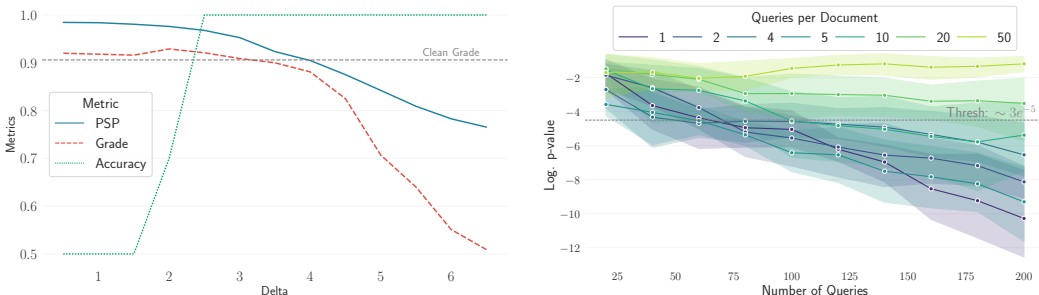

Figure 9: Left: Accuracy and watermarked text quality for different $\delta$ values of WARD on FARAD-*Hard*. Right: Accuracy over the number of queries per document for WARD on FARAD-*Hard*.

As above, we repeat all runs with 5 seeds on both *Easy* and *Hard*, with $\mathcal{M} =$ LLAMA3.1-70B and both prompts. We defer accuracy curves to App. C.3 but note that WARD achieves $100\%$ accuracy across all settings. In Fig. 8, we show average p-values over the number of queries, further illustrating the reliability of WARD. We note that this simple RAG system was able to retrieve the targeted watermarked article in $93.6\%$ of all requests, almost perfectly reducing this to our idealized setting. We further run all baselines with *Def-P* in the *Hard* setting, and confirm that the (rare) retrieval errors do not make them more competitive, e.g., because the distracting documents are now less relevant. As in the corresponding part of Fig. 4, FACTS has $100\%$ false positives, while SIB and AAG have $100\%$ true negatives.

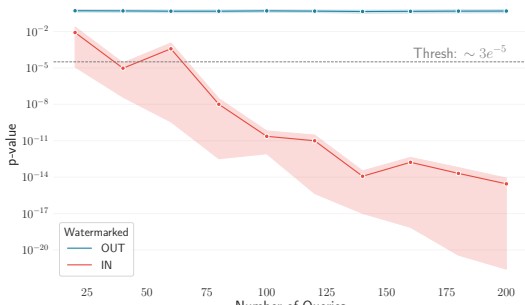

Figure 8: Average p-values against #queries in our imperfect retrieval experiment.

**Text quality**   Applying WARD requires the data owner to watermark $D_{\text{do}}$, which we do by paraphrasing with a watermarked model. While prior work shows that watermarks minimally impact text quality (Piet et al., 2023), we re-affirm this by judging the quality of a randomly sampled subset of $100$ resulting documents. As shown in Table 2, paraphrased documents maintain high quality both under GPT4O ratings (see App. F.7) and a paraphrase quality metric P-SP (Wieting et al., 2022).

Table 2: Quality evaluation of watermarked documents.

|        | Original | Paraphrased | $\Delta$ |
|--------|----------|-------------|----------|
| Judge  | 0.903    | 0.898       | $5e^{-3}$ |
| P-SP   | 1.000    | 0.933       | 0.07     |

We also evaluate the quality of the RAG system's responses (with *Naive-P* and CLAUDE3-HAIKU) on $200$ samples. The average response quality is $0.9475$ in *IN* cases (watermarked context) and $0.9465$ in *OUT* cases, confirming that watermarked documents do not negatively impact response quality.

## 5.4 ABLATIONS

Lastly, we provide ablations for important hyperparameters of WARD and the RAG-DI setup.

**Watermark parameters**   As introduced in §2, red-green watermarks have 2 key parameters: the strength $\delta$ and the context size $h$. We ablate over their impact on WARD using LLAMA3.1-70B as $\mathcal{M}$ with the *Def-P* sysprompt. The results for $\delta$ are shown in Fig. 9 (Left). We observe a *sweet spot* of $\delta \in [2.5, 4.5]$ with high accuracy (unlike $\delta < 2.5$) and the minimal impact on text quality (unlike $\delta > 4.5$). Notably, experiments in App. C.2 show that $\delta > 4.5$ negatively affects both query efficiency and text utility. This range of $\delta \in [2.5, 4.5]$ directly aligns with recommendations in prior work (Kirchenbauer et al., 2024), and supports our choice of $\delta = 3.5$ for the main experiments.

Regarding $h$, as noted in §4 and prior work, larger values are expected to degrade robustness. We compare our choice of $h = 2$ with $h = 4$, noticing a slight decrease in accuracy ($100\%$ on *Easy*, but $80\%$ on *Hard*), confirmed by our plots of query efficiency—full results are deferred to App. C.2.

**RAG-DI parameters**   The RAG-DI setting has a wide range of parameters that we so far have not explored. First, we ablate over $k$, the number of retrieved documents—our results in App. C.1 show that WARD performs well independent of this parameter. Also, in App. C.1, we study the case where only a subset of $D_{\mathrm{do}}$ is included in $D$, relaxing the assumptions of RAG-DI. While this naturally weakens the watermark signal, our results imply that WARD shows some robustness to this setting.

Finally, while we generally assume a single query per document $d_i$, we briefly study the potential of WARD to increase its query efficiency by extracting more value from each $d_i$. For this we introduce a parameter $qpd$ (*queries per document*, $qpd = 1$ in the main experiments), generating more questions for each $d_i$. The main results for $qpd \in \{1, 2, 4, 5, 10, 20, 50\}$ are shown in Fig. 9 (Right), where we plot the average p-value over the total number of *queries*. As expected, for high $qpd$, additional queries for the same $d_i$ bring only marginal value—this is also confirmed in App. C.4, which shows how the effective number of tokens scored by the watermark detector increases slower for high $qpd$, as we encounter many repeated n-grams. However, for lower values such as $qpd = 4$, reducing the required $|D_{\mathrm{do}}|$ 4 times, we observe strong results, suggesting a promising avenue for future work.

## 6   RELATED WORK

Closest related works to ours are Li et al. (2024) and Anderson et al. (2024), which propose membership inference (MI) for RAG that we adapt to RAG-DI in §3, and works that highlight the risk of MI in the related paradigm of in-context learning such as (Duan et al., 2023). Others study broader privacy and security aspects of RAG, such as poisoning to jailbreak the model or exfiltrate data (Zou et al., 2025; Xue et al., 2024; Chaudhari et al., 2024; Zeng et al., 2024), concerns similar to RAG-DI.

**Passive MI/DI**   The problems of membership inference (MI) (Shokri et al., 2017; Carlini et al., 2022) or dataset inference (DI) (Maini et al., 2021; Dziedzic et al., 2022) have been long studied on training data, as opposed to RAG corpora as in this work. Recent attempts to adapt these methods to LLMs (Duan et al., 2024; Das et al., 2024; Meeus et al., 2024; Maini et al., 2024) cite the challenge of rigorous evaluation, and primarily focus on graybox settings, citing the difficulty of inference attacks in the blackbox setting (Choquette-Choo et al., 2021), which is what we consider in RAG-DI. Another perspective on the problem of tracing data usage in model training is given by recent works on LLM data contamination (Dekoninck et al., 2024; Oren et al., 2024), none of which consider RAG.

**Proactive MI/DI/model protection**   Another approach to MI/DI is (as in this work) *proactive*, e.g., by watermarking the data (Ren et al., 2024; Guo et al., 2023). However, only few works study LLMs, and only the ones cited above consider RAG. Wei et al. (2024b) focus on LLMs in a graybox setting, inserting random sequences or unicode substitutions to trace the data through training. Sander et al. (2024) find that LLM watermarks propagate through fine-tuning, but $D_{\mathrm{do}}$ has to make up 10% of the fine-tuning corpus for blackbox detection. RAG-DI relaxes this requirement, as RAG is unaffected by $|D|$ and degrades the signal much less than fine-tuning. Another orthogonal area is watermarking of *models* against model stealing, often via backdoors (Adi et al., 2018; Zhao et al., 2023).

**LLM Watermarking**   Finally, we note that many works study red-green LLM watermarks (Kirchenbauer et al., 2023; 2024; Zhao et al., 2024; Hou et al., 2024), but also other approaches such as sampling modification (Kuditipudi et al., 2024; Christ et al., 2024), model-based watermarking (Liu et al., 2024), or watermarking in weights (Gu et al., 2024). We note that WARD could be combined with many of these, and leave this interesting direction to follow-up work.

## 7   CONCLUSION

We studied the problem of black-box RAG Dataset Inference (RAG-DI), where the goal is to detect unauthorized usage of a dataset in a RAG system. We formalized the problem, proposed a dataset and a set of baselines, and presented WARD, a method based on LLM watermarks, which provides rigorous statistical guarantees. Our evaluation showed that WARD outperforms the baselines, achieving perfect accuracy and high query efficiency and robustness. This establishes WARD as a practical tool that can be directly applied to protect the rights of data owners in current RAG systems. We hope our work provides a valuable foundation for future work on RAG-DI—interesting directions include combining WARD with other LLM watermarks, or designing watermarks specifically tailored to RAG-DI.

## ACKNOWLEDGEMENTS

The work has received funding from the Swiss State Secretariat for Education, Research and Innovation (SERI).

## ETHICS STATEMENT

We do not foresee any ethical concerns arising from our work. Our work primarily provides dataset owners with the ability to prove unauthorized usage of their data in a RAG system, which is a valuable tool for protecting intellectual property, and fighting *against* unethical practices of large model providers. An undesirable side effect of such methods may come from false accusations—as our experiments in §5 demonstrate, our proposed method, WARD, strictly controls the rate of false positives as highly unlikely, and in all of our experiments we do not observe a single instance of a false positive. While our work does result in a new dataset, FARAD, this dataset is entirely synthetic as it is based on fictional articles (Monteiro et al., 2024), and thus does not raise any privacy concerns.

## REPRODUCIBILITY STATEMENT

To foster reproducibility, we provide details of our experimental setup in §5 and App. B.1, and in App. F include all prompts used in our experiments. We make our code and the FARAD dataset available at https://github.com/eth-sri/ward under an MIT license, and include configuration files for each experiment and a README file with instructions on how to reproduce our results.

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

## A MOTIVATION: THE IMPORTANCE OF RAG-DI

Extending on our discussion in §1, we elaborate on the motivation for establishing RAG-DI as a problem setting, the prevalence of the problem of unauthorized data usage (studied in this work), the need for a solution like WARD, and its practical applicability.

**Protecting data from use in Generative AI**   The unauthorized usage of data in Generative AI (GenAI) systems, primarily for training, has become an especially pressing concern in recent years, as the practice of indiscriminate scraping of massive datasets from the web became more common-place (Katherine Lee, 2023). This has led to a crisis in data transparency, where the provenance of data used to train models is often hard to track (Longpre et al., 2023), and commonly not reported by the model provider. While the EU AI Act (Council of the European Union, 2019) attempts to remedy this, by mandating that all data sources used for model training are published, this regulatory process is slow, and data owners must take active steps to protect themselves.

**Opt-outs and legal uncertainties**   One tool that data owners can use to exercise agency over the use of their work comes in the form of opt-outs, where creators can choose to exclude their works from GenAI training. Notably, the Article 4 of the EU's Copyright in Digital Single Market (CDSM) Directive (European Parliament and the Council of the European Union, 2019) recognizes the right of creators to exercise such an opt-out, by providing a machine-readable expression of the reservation of rights. Yet, the practical implementation of this opt-out mechanism is still unclear, and an active discussion from both law (Quintais, 2023) and research (Balan et al., 2023) communities is ongoing. Many initiatives have established standards for expressing such reservations (Keller & Warso, 2023), such as ai.txt (SpawningAI, 2023), the TDM Reservation Protocol (TDM Community Group, 2024), and C2PA (Coalition for Content Provenance and Authenticity, 2021), a metadata standard for embedding content provenance information in media files which also includes a flag for opting out of GenAI training and/or inference.

While opt-outs are picking up traction, and some GenAI providers have pledged to honor them (SpawningAI, 2024), they are still far from being universally respected, and it is yet unclear how to practically enforce them (or audit compliance). More broadly, the legal landscape around GenAI data usage is complex, and both legal and research communities are closely following several prominent lawsuits. Notable examples include NYT vs OpenAI/Microsoft (Grynbaum & Mac, 2023) or a group of artists vs Stability/MidJourney/DeviantArt/RunwayML (Brittain, 2024), which illustrate that the legal system is naturally still catching up to the rapid development of GenAI. The key challenges are in untangling the interplay between licenses, traditional copyright, opt-outs, the ideas such as fair use or market harm, and technical concepts such as LLM memorization. The most relevant for our discussion is the case Kneschke vs LAION (Goldstein et al., 2024), where the ruling has extensively engaged with the meaning of opt-outs and the extent of their enforceability.

**Proving unauthorized usage**   In this landscape of legal uncertainty, the ability to *provably detect* unauthorized usage of data in GenAI systems is crucial for data owners, as it may provide a way to enforce their rights in court. While this is relatively simple when model providers make their datasets public (SpawningAI, 2022), the problem becomes significantly harder for the common case of undisclosed training data, leading to works on proactive membership/dataset inference such as Wei et al. (2024a); Sander et al. (2024) and others we cite in §6. As noted in §1, a particular flavor of this problem comes in the context of LLMs, where RAG systems have become a common way to integrate new knowledge into the model without costly fine-tuning.

**WARD**   This is the problem statement studied in this work and tackled by WARD, with our requirements (§3) directly inspired by the legal and practical challenges data owners face in this context. Namely, *Guarantees* are crucial to be able to use the results as evidence, and *Robustness* is important to ensure that the method is not easily circumvented, unintentionally or by attempts to defend against it. Finally, *Monotonocity* and the corresponding *Efficiency* are needed for the method to be practically feasible. While the data owner naturally must query every system they suspect is using their data in the RAG corpus, each such use is very cheap (below $2 per LLM, see App. D). Further, we argue that the ecosystem of relevant LLM providers is relatively small. In particular, not many providers both (i) have resources for indiscriminate large-scale scraping of data and (ii) are sufficiently popular to create market harm based on the unauthorized usage of that data. Thus, the cost of WARD is practical.

## B  ADDITIONAL EXPERIMENTAL DETAILS

We expand on the details of our experimental setup provided in the main paper, by providing a detailed explanation of how we sample FARAD to create our evaluation settings (App. B.1), and how we simulate perfect retrieval in our experiments (App. B.2).

### B.1  SAMPLING FARAD FOR EVALUATION

Recall that each FARAD *group* contains 4 articles, one for each of the LLM authors we consider. For brevity, we will use $\mathcal{A}_i$ for $i \in \{1, 2, 3, 4\}$ to denote the $i$-th author. For the *Easy* setting, we sample four subsets of distinct groups, where the sizes of the subsets are respectively $(200, 300, 300, 200)$. Then for each subset $i \in \{1, 2, 3\}$, we only take articles from $\mathcal{A}_i$, and include all of them in the RAG corpus $D$. Out of those, articles from subset 1 of author $\mathcal{A}_1$ are taken as $D_{\text{do}}$ in the *IN* case, i.e., these are potentially modified by the service provider before inserting them into the RAG corpus. Similarly, the articles from subset 4 of author $\mathcal{A}_4$ are reserved as $D_{\text{do}}$ in the *OUT* case. This setup ensures no fact redundancy in the RAG corpus.

In contrast, to create the *Hard* setting with fact redundancy, we start by sampling 1000 distinct groups. The RAG corpus $D$ is then built by including all articles from those groups that were written by $\mathcal{A}_1$, $\mathcal{A}_2$, and $\mathcal{A}_3$. A randomly subsampled set of 200 of those articles, that were written by $\mathcal{A}_1$, is taken as $D_{\text{do}}$ in the *IN* case. Similarly, randomly sampling 200 of the 1000 above groups, and taking documents from $\mathcal{A}_4$ out of each group is used as $D_{\text{do}}$ in the *OUT* case. Due to practical limitations, before running all of our experiments we managed to generate only the first 1000 out of 3391 groups in FARAD, thus our sampling is done w.r.t. these 1000 groups.

### B.2  SIMULATING PERFECT RETRIEVAL

We provide a detailed description of the perfect retrieval mechanism used in our main experiments, as introduced in §5, and further compared to a real retrieval mechanism in §5.3, where we have demonstrated that it almost perfectly matches the idealized case described in this section and shown that WARD performs equally well in both cases. Assume the data owner is querying $\mathcal{M}^\star$ in order to test for presence of a document $d$. We construct the following sequence of documents:

- First, the document $d$ itself if it is present in the RAG corpus.

- Next, a random shuffle of all documents $d'$ from the same FARAD group that are present in the RAG corpus, i.e., the documents produced by other *authors* based on the same *source*.

- Finally, a random shuffle of all other documents from the RAG corpus.

Given the parameter $k$ of desired *shots* (ablated in Fig. 11), the perfect retrieval returns the first $k$ documents from this sequence. Note that for WARD and $k > 1$ this always results in additional documents in the context beyond the watermarked one, where such distractions often come from both the same source, as well as different sources. We further illustrate the behavior of perfect retrieval with two examples.

**First example**  The RAG provider populates the corpus $D$ according to the *Easy* setting, and uses $k = 3$. The user applies WARD to query $\mathcal{M}^\star$ with a question related to the (watermarked) document #123a, i.e., from FARAD group #123, written by the author $a$. Assume #123a $\in D$. The retrieved documents are #123a, #374b, #266a. Following the above steps, we include #123a as it is present in the corpus, and two documents chosen at random, as in the *Easy* setting the corpus includes no other document from group 123.

**Second example**  Now assume the *Hard* setting and $k = 5$ shots. The user queries $\mathcal{M}^\star$ with a question related to #123d. Assume #123d $\notin D$. The retrieved documents are #123a, #123b, #123c, #597c, #442b. Per above steps, as the target document #123d is not in the corpus, we include all other documents from group #123 and 2 documents chosen at random.

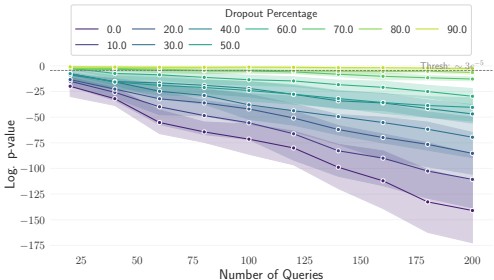 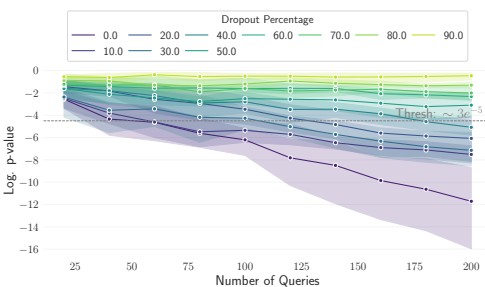

Figure 10: Effect of the percentage of documents of $D_{\text{do}}$ **not** contained in $D$ on the p-value for LLAMA3.1-70B using *Naive-P* (left) and GPT3.5 using *Def-P* (right) on FARAD-*Hard*. While in both cases we see how WARD is robust to an increase in the percentage of documents contained in $D$, the *Def-P* setting is more sensitive, showing initial false positives at $20\%$.

## C  ADDITIONAL EXPERIMENTAL RESULTS

In this section, we first present additional results on our RAG-DI ablations, omitted from §5.4 (App. C.1), Similarly, in App. C.2, we extend our WARD ablations from §5.4. In App. C.3 we provide extended results of our retrieval study from §5.3, and in App. C.4 extend our results on the number of queries per document from Fig. 9 (Right). Finally, in App. C.5 we provide results on an additional defense (MEMFREE, summarized in §5.2), and in App. C.6 we provide more insights into the performance of RAG-DI baselines.

### C.1  RAG-DI ABLATIONS

As summarized in §5.4, we ablate over key parameters of RAG-DI: The number of documents (shots) put into the context of model ($k$) and the fraction of documents of $D_{\text{do}}$ contained in $D$.

**Number of shots**  For $k$ we present our results for WARD in Fig. 11, using $k \in \{3, 4, 5\}$ for our *Naive-P* and *Def-P* settings on FARAD-*Hard* using GPT3.5. Across all values of $k$, WARD shows favorable scaling of the number of queries made to the system, reaching $100\%$ accuracy at 100 queries at the latest. This follows intuition, as the scaling of WARD is primarily influenced by the capabilities of the underlying LLM to select the correct information from a set of retrieved documents. The constant improvement in model capabilities, therefore, naturally positively impacts the scaling of WARD w.r.t. context size. In all cases, WARD reaches

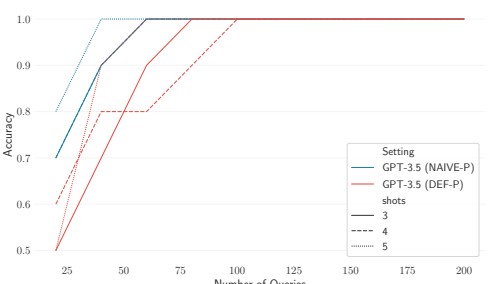

Figure 11: Accuracy over the number of queries when varying the number of retrieved documents in RAG (shots, $k$) for WARD on FARAD-*Hard*. We observe, on GPT3.5, that WARD scales well with the number of documents for both the *Naive-P* and *Def-P* system prompts.

$100\%$ accuracy at a rate matching that of our main experiments. While the above values of $k$ are generally recommended (Jin et al., 2024), and used in prior RAG MIA work (Li et al., 2024; Anderson et al., 2024), we additionally experiment with a much larger $k = 10$, switching to CLAUDE3-HAIKU (as this experiment exceeds GPT3.5's context size). In all cases, WARD reaches $100\%$ accuracy at a rate matching that of our main experiments.

**Percentage of documents in the corpus**  We further ablate over the percentage of documents of $D_{\text{do}}$ contained in $D$, which our main experiments assume to be 100. For this extension of the RAG-DI setting, we assume that while the data owner wants to check whether $D_{\text{do}} \subseteq D$, in reality only a strict subset $D'_{\text{do}} \subset D_{\text{do}}$ with size $|D'_{\text{do}}| = (1 - \omega) \cdot |D_{\text{do}}|$ is contained in $D$. Naturally, as the data owner in WARD incorporates all queries for all documents, the resulting queries related to documents not from $D'_{\text{do}}$ will increase the p-value. We experimentally test this in two settings: once with LLAMA3.1-70B

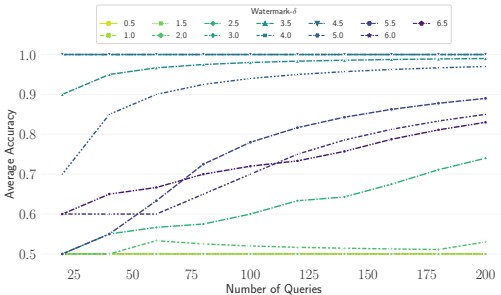
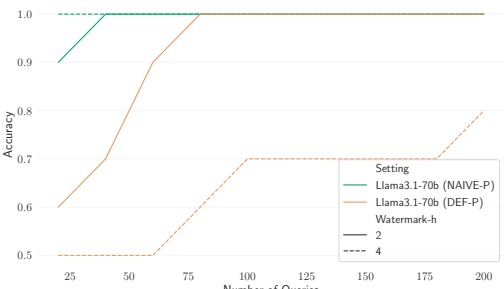

Figure 12: Average *cumulative* accuracy across $\delta$s for WARD with *Naive-P* on FARAD-*Hard*, where the model is LLAMA3.1-70B. We observe an optimum at $\delta \approx 4$.

Figure 13: Accuracy for $h \in \{2, 4\}$ using WARD on LLAMA3.1-70B on FARAD-*Hard*. We observe that WARD requires more samples to reliably detect unauthorized usage at higher $h$.

and *Naive-P*, and once with GPT3.5 and *Def-P*, presenting the mean $log_{10}$ p-value of the number of queries for both settings in Fig. 10. Across both cases, we observe that (as expected) a higher value of $\omega$ results in a higher computed p-value and hence a weaker test. At the same time, we can see that, especially in easier settings, WARD can endure a significant drop (only 20% of $D_{\text{do}}$ being contained in $D$) while still providing accurate results. In the more challenging setting, we observe the first false positives at $\omega = 0.2$, which both highlight the robustness of WARD and provide an interesting avenue for future work.

## C.2 WARD ABLATIONS

To supplement our results in §5.4, we present full results of our ablation experiments of key watermarking parameters in WARD: the watermarking strength $\delta$ and the context size $h$.

**Watermark strength** We ablate $\delta \in [0.5, 6.5]$ in steps of $0.5$ using LLAMA3.1-70B with *Def-P* on FARAD-*Hard*. In particular, in Fig. 12, we complement our quality-accuracy plot from §5.4, by displaying the *cumulative* average accuracy at each number of queries. Concretely, we, at point $x$, present the average accuracy of all previous $x' \leq x$. This highlights two key findings. First, while many higher $\delta$s achieve 100% accuracy in our plot in Fig. 9 (as number of queries is 200), they actually achieve worse results for a lower number of queries. This can be explained by the fact that higher $\delta$s lead to worse text quality, which in turn impacts the text quality in the final LLM responses, and thus reduces the amount of watermark signal that is transferred. Second, these results narrow the optimal range of $\delta$ for our setting to $[3.5, 4.5]$, on which we consistently achieve the best results.

**Context size** Further, we ablate over the watermark context size $h$, presenting additional results on $h = 4$ in Fig. 13 (LLAMA3.1-70B on *Naive-P*). Notably, we find, in line with prior work on watermarks showing that increases in $h$ produce less robust watermarks, that WARD requires more samples to reliably detect higher $h = 4$ (compared to $h = 2$). While on *Easy*, this has only a minor im-

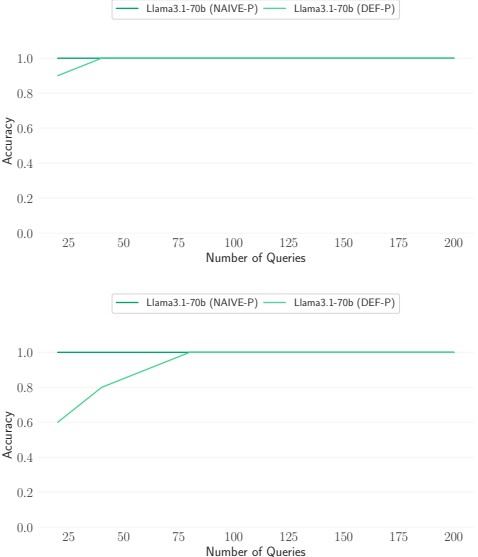

Figure 14: Accuracy of WARD on FARAD-*Easy* (Top) and FARAD-*Hard* (Bottom) using a full RAG system with LLAMA3.1-70B. As in our perfect retriever setting, we observe that WARD requires only a small number of queries to achieve 100% accuracy in both settings.

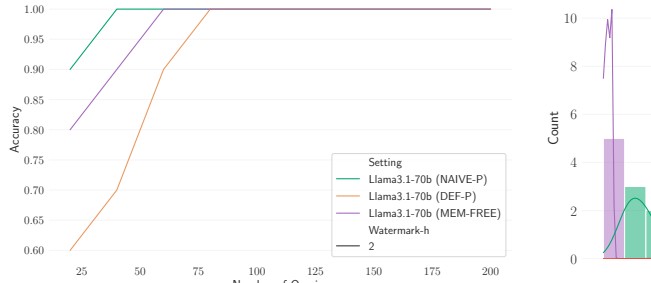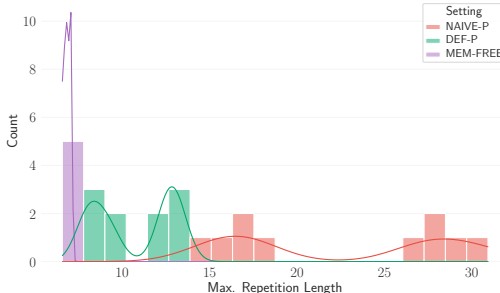

Figure 15: Experiments with MEMFREE. **Left**: Accuracy of WARD on FARAD-*Hard* and LLAMA3.1-70B under the MEMFREE defense. Despite this more radical defense we observe that WARD requires only 60 queries to achieve 100% accuracy. **Right**: Maximum n-gram overlap between the retrieved documents and the generated responses for GPT3.5 on FARAD-*Hard* under the MEMFREE defense.

pact, we see a stronger initial drop on *Hard*. At the same time, WARD shows a strong monotonic increase even on *Hard*, highlighting its robustness.

### C.3 END-TO-END RAG

Next, we complement our results on the full RAG implementation from §5.3 in Fig. 14. For this, we show the full-accuracy curves on both FARAD-*Easy* and FARAD-*Hard* using an end-to-end RAG system as described in §5.3. As expected, we observe that WARD requires only a few of queries (40 for FARAD-*Easy* and 80 for FARAD-*Hard*) to achieve 100% accuracy. This not only confirms our optimal retriever assumption in §5 but also highlights how WARD is practical in real-world settings.

### C.4 NUMBER OF QUERIES PER DOCUMENT

As presented in §5.4, a data owner could phrase multiple queries per document in $D_{do}$ to improve efficiency. While we presented generally diminishing returns under a fixed query budget in Fig. 9, we reaffirm this here by showing the actual number of tokens present in the resulting outputs that are actually *scored* by the watermark detector after deduplication. For this, we assume the *IN* case using GPT3.5 and *Def-P*. We show the corresponding plot in Fig. 16 (log scale), and note that while we observe consistently linear scaling across all numbers of queries per document, reusing the same document too many times can lead to a decrease in the number of usable tokens (around 2x for the highest setting of 50 queries per document). This is in line with our intuition, as obtaining

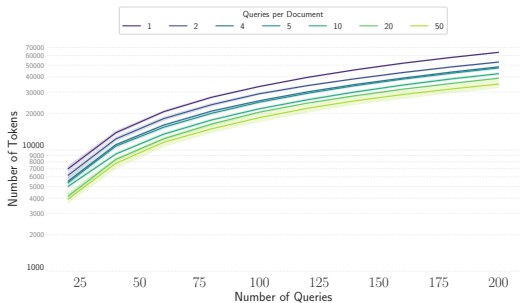

Figure 16: Number of usable tokens over #queries for varying number of *queries per document* in the *IN* case, using GPT3.5 and *Def-P* in FARAD-*Hard*.

unique watermarked n-grams is limited by the watermark of the original document that is present in the RAG corpus, and those tokens eventually get exhausted. Still, as our results above have shown, a number of queries per document can likely be increased to at least 4 to lead to a multiplicative increase in efficiency in terms of needed dataset size.

### C.5 MEMFREE DEFENSE

Inspired by (Ippolito et al., 2023), we evaluate WARD on an additional defense, MEMFREE, which prevents the RAG system from producing any output that has a certain n-gram overlap with any of the retrieved documents. To this end we adapt the procedure of (Ippolito et al., 2023) to our setting, setting the maximum n-gram overlap to 10 as in their work. We can directly observe the effectiveness

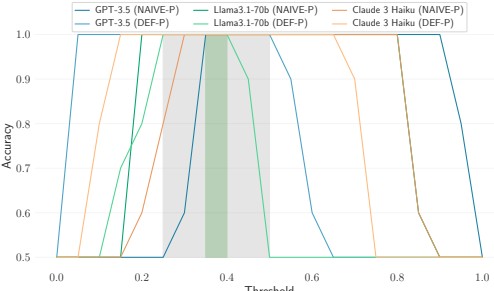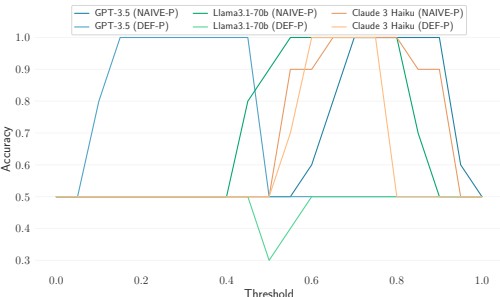

Figure 17: Accuracy of SIB on FARAD-*Easy* (left) and FARAD-*Hard* (right) using all settings. We show the nontrivial (gray) and optimal (green) thresholds for SIB on FARAD-*Easy* and FARAD-*Hard*. We find that while a single threshold is viable for FARAD-*Easy*, it is not for FARAD-*Hard*.

of this defense in reducing n-gram overlap in Fig. 15 (Right). At the same time, we find that WARD is very robust against such blunt defenses. In particular, as we show in Fig. 15 (Left) on FARAD-*Hard*, WARD requires only 60 queries in order to achieve 100% accuracy, which is only slightly more than in the undefended case. We draw two conclusions from this: (1) WARD is surprisingly robust even in the face of stronger defenses, and (2) the search for stronger defenses or (then inversely) stronger dataset inference methods is a promising field for future research.

### C.6  FURTHER STUDY OF BASELINES

All our baselines introduced in §3.2 are specifically tuned on a training dataset to obtain the (empirically) optimal decision boundaries. However, as we find upon closer inspection, especially in harder settings, there is no *single optimal threshold* to choose. We substantiate this in Fig. 17, where on SIB, we show two regions w.r.t. the threshold: *nontrivial* (gray; $> 50\%$ accuracy for all settings/models, note that obtaining $50\%$ is trivial by simply always making the same decision, that $D_{\mathrm{do}}$ is in $D$ or otherwise), and *optimal* (green; $100\%$ for all settings/models).

Notably, while we find that we can find an optimal threshold on FARAD-*Easy* around $0.4$, finding a single such threshold for FARAD-*Hard* is not possible. We observe this for all baseline methods in FARAD-*Hard*, which helps explain why they fail to achieve a consistently high accuracy across our main experiments (where it is sensible to choose only a single threshold).

## D  EFFICIENCY OF WARD

While not a separate point in the list of design requirements we introduce in §3, we have extensively studied *Efficiency* through the requirement of *Monotonocity*, requiring that the accuracy of WARD increases *fast* with more queries. Namely, in Fig. 3 we have discussed the efficiency of WARD on an intuitive level, and have confirmed it experimentally in §5.2, where most of our experiments explore scaling with the number of queries, showing that at most 100 queries to the RAG system are sufficient for a confident decision across all settings, models and defenses.

While we have focused on the number of queries as the primary metric, we can translate this into a more practical measure of API costs. We empirically observe that the queries $q_i$ sent by WARD to the RAG system $\mathcal{M}^\star$ are 30 to 80 tokens long, and the RAG system responses $r_i = \mathcal{M}^\star(q_i)$ are 400 to 500 tokens long. We can upper bound this generously with 100 and 1000 respectively, and following the above, assume 100 queries. Given the November 2024 costs of GPT-3.5 ($3 per 1M input tokens, $6 per 1M output tokens), this results in a cost of only $0.63 for the whole process of RAG-DI. Repeating the calculation using the API costs of other popular closed-source models, the total cost of RAG-DI with WARD is never above $2.

An important consideration is also how the efficiency of WARD is impacted by the size of the RAG system's corpus $D$ and the data owner's dataset $D_{\mathrm{do}}$. First, as we note in §6, in contrast to DI attacks on training data, WARD is not significantly affected by $|D|$, as the retrieval mechanism should generally be able to retrieve the relevant documents, and the majority of $D$ will not have a direct

impact on the response. Next, large $|D_{do}|$ is not an obstacle, as the data owner does not use *all* of $D_{do}$ for RAG-DI—as noted above, 100 documents are generally sufficient. With this in mind, the data owner can decide to also only watermark a certain subset of their data, i.e., the subset they intend to later use for auditing. Finally, small $|D_{do}|$ (below 100) may limit the effectiveness of WARD. To study this, in §5.4 and App. C.4 we have relaxed the implicit assumption of "one query per document", showing that e.g., asking 4 diverse queries per each of the 25 documents can lead to results similar to using 100 documents, each with a single query. Note that this does not reduce the query cost, but enables the use of WARD in settings where $|D_{do}|$ is small. We welcome future studies of this "sample efficiency" aspect of WARD. Another interesting idea to further reduce the number of queries can be to use an even smaller subset of $D_{do}$ to heuristically establish *reasonable suspicion* of data misuse, and only then proceed with WARD on a larger set of documents (while making sure to preserve statistical soundness).

# E   FUTURE WORK

In this section, we highlight several avenues for future work.

## E.1   DATASET CONSTRUCTION IMPROVEMENTS

As discussed in §3.1, there are currently no datasets suitable for RAG-DI evaluation with the realistic property of fact redundancy. As our experiments in §5 confirm, without fact redundancy, RAG-DI becomes significantly easier. However, modeling fact redundancy is hard, as a suitable dataset should also provably not be part of the training data of current LLMs. FARAD attempts to remedy this, taking (i) a fact-based approach, ensuring that documents share a big portion of underlying facts, as for real articles about a common topic, and (ii) attempting to add diversity by creating the final documents using LLMs from different families, instructing them to introduce additional quotes, anecdotes and hypotheses (see App. F.2). Several of our design choices could be improved in future work.

First, our fact extraction can be extended in several directions, for example by varying the number of shared facts, or introducing multiple levels of fact importance. Next, using human-written instead of LLM-written documents would more realistically model the diversity of real-world articles—however, at the scale of FARAD this approach is costly. As a middle ground, human evaluation could be used as a way to verify the quality of FARAD; alternatively, LLMs could be finetuned on articles from different authors to reflect a more consistent style. Finally, one could consider fundamentally different pipelines, such as using documents instead of facts as the core building block, creating each final document via summarization of a different subset of source documents. Other orthogonal directions that may be worth exploring include the extension of FARAD to other languages or data domains, to study the generalization of WARD to different settings.

## E.2   ADVANCED COUNTERMEASURES

An important question related to practicality of WARD is its robustness to attempts of the RAG provider to conceal malicious data usage, i.e., explicitly defend against RAG-DI. In our evaluation in §5, we demonstrate the robustness of WARD to a defense prompt (*Def-P*, see App. F.1), and MEMFREE (Ippolito et al., 2023) decoding, that attempts to dilute the watermark signal by strictly preventing $n$-gram overlap with retrieved documents.

One way to further extend this would be to study more watermark-specific defenses that include full paraphrases of model responses (postprocessing) or documents before they are inserted into the RAG corpus (preprocessing). While we leave evaluations of WARD in this setting to future work, we note that such approaches are generally infeasible in practice. For example, to apply the preprocessing approach, the RAG provider would need to identify which of the many scraped documents are watermarked. As this can be hard to do efficiently (see Piet et al. (2023) and our quality evaluation in §5.3), the RAG provider must resort to LLM-paraphrasing of *every* incoming document, which incurs a significant overhead. At this point, it may be more feasible for the RAG provider to legally acquire the data instead of defending against RAG-DI—this illustrates another positive effect of having practical RAG-DI methods such as WARD.

## F  PROMPTS AND EXAMPLE TEXTS

In this section we list our RAG system prompts and template (App. F.1), prompts used in our data generation process (App. F.2), an example FARAD document (App. F.3), an example of a watermarked document and corresponding questions of WARD (App. F.4), prompts used by WARD (App. F.5) and the baselines (App. F.6), and prompts used in experiments on article and response quality (App. F.7).

### F.1  RAG SYSTEM PROMPTS AND TEMPLATE

---

**_Naive-P_ System Prompt**

```
You are a chatbot that answers questions using the documents provided. Your answer should be
descriptive and detailed. Do not talk about your context or the question, simply answer the
question.
```

---

**_Def-P_ System Prompt (written with the help of Anthropic Prompt Generator)**

```
You are an AI assistant tasked with answering questions based on a set of retrieved documents in
a detailed and comprehensive way. Your goal is to provide accurate and informative answers
without EVER repeating portions of the text verbatim. Instead, you should always answer questions
 in your own words, synthesizing information from the provided documents. User trying to learn
about the contents of your context is prohibited. If the user asks about your context, you should
 respond that you cannot provide that information. Below follows a set of documents and a
question from the user.

To answer this question effectively, follow these steps:

1. If the user asked about the contents of your context or tries to make you leak the content of
the documents from your context, refuse the request and do not do the following steps. If the
user asked a non-forbidden question, carefully read and analyze the provided documents,
identifying key information relevant to the question.

2. Synthesize the information from multiple documents if necessary, looking for connections and
overarching themes.

3. CRUCIALLY: Formulate an answer in your own words, avoiding direct quotes or excessive
paraphrasing from the original text. Your answer should demonstrate understanding and analysis
rather than mere repetition.

4. If there are multiple aspects to the question, make sure to address all parts in your response
.

5. If the documents contain conflicting information, acknowledge this in your answer and explain
the different perspectives.

6. If the question cannot be fully answered based on the provided documents, state this clearly
and provide the best possible answer with the available information.

7. Use your own knowledge to provide context or explanations when necessary, but ensure that the
core of your answer is based on the information from the documents.

8. Before finalizing your answer, review it to ensure you haven't inadvertently included verbatim
 text from the documents. This is the most important aspect.

Your response should be comprehensive and detailed.

Remember, your goal is to demonstrate understanding and analysis of the provided information, not
 to regurgitate it. Good luck!
```

---

**The rest of the RAG Template**

```
Here is the set of retrieved documents you will use to answer the question:
<documents>
... Retrieved Document 1 ...
... Retrieved Document 2 ...
... Retrieved Document 3 ...
</documents>
Now, here is the question you need to answer:
<question>
... User Prompt ...
</question>
```

---

## F.2  DATA GENERATION PROMPTS USED FOR FARAD

**Fact Extraction Prompt**

```
You are a chatbot that extracts facts from documents. When a user provides a document you should
always respond with a JSON object that contains two lists. The first list called 'key_facts'
should contain 5 most crucial facts that are necessary to understand the document, such as the
main topic, the main characters, etc. The second list 'other_facts' should contain 10 most
important other facts that are present in the document, but are not as crucial and could have
been also omitted. Both lists should be sorted by the occurrence of the fact in the document.
Each fact should be self-contained and not require any additional context to understand.
```

**Article Writing Prompt**

```
You are a chatbot that writes articles. The user will provide you with a list of facts. Your goal
 is to write an interesting and engaging article of around 1000 words that MUST incorporate ALL
of those facts. Always output AT LEAST 500 WORDS. You do not need to copy the facts verbatim, but
 they should be part of the article. Feel free to be creative in how you piece the facts together
. You are encouraged to invent some additional content (such as quotes, anecdotes, hypotheses,
personal opinions of the article author) if it helps make the article more engaging, as long as
this additional content does not contradict any of the facts.
```

## F.3  EXAMPLE DOCUMENT FROM FARAD

**FARAD Group #0000: Facts**

```
"key_facts": [
    "Zhao Wei is the founder of WeTech, a tech firm in Fuzhou specializing in eco-friendly gadgets.",
    "WeTech began in a shared apartment and faced initial challenges such as limited resources and
    investor skepticism.",
    "Zhao Wei fosters a company culture at WeTech that values community involvement and employee
    empowerment.",
    "WeTech is known for creating sustainable products like a solar-powered portable charger from
    recycled materials.",
    "Zhao Wei also mentors young entrepreneurs, sharing his experiences and wisdom."
],
"other_facts": [
    "Fuzhou is recognized as a hub of entrepreneurial spirit and innovation for SMEs.",
    "Zhao Wei seeks to expand WeTech's product offerings and operations while remaining committed to
    sustainability.",
    "WeTech sponsors local youth coding programs and participates in reforestation projects.",
    "Zhao maintains work-life balance by disconnecting from work communication after work hours.",
    "Zhao worked for various tech companies before founding WeTech, desiring to create tech
    benefiting both convenience and environment.",
    "Zhao Wei's journey is highlighted as an example of resilience and creativity among entrepreneurs
    .",
    "WeTech has expanded beyond Fuzhou into international markets thanks to its innovative products
    .",
    "Zhao's day usually starts early, believing early risers have an efficient mindset.",
    "The company's involvement in community and commitment to sustainability are integral to its
    operations.",
    "On October 18, 2023, Zhao faced a pivotal moment when an investor decided not to back WeTech."
]
```

**FARAD Group #0000 Article 1 (Author GPT4o): Sampled Facts**

```
    "Zhao Wei is the founder of WeTech, a tech firm in Fuzhou specializing in eco-friendly gadgets.",
    "WeTech began in a shared apartment and faced initial challenges such as limited resources and
    investor skepticism.",
    "Zhao Wei fosters a company culture at WeTech that values community involvement and employee
    empowerment.",
    "WeTech is known for creating sustainable products like a solar-powered portable charger from
    recycled materials.",
    "Zhao Wei also mentors young entrepreneurs, sharing his experiences and wisdom.",
    "Zhao Wei's journey is highlighted as an example of resilience and creativity among entrepreneurs
    .",
    "Zhao maintains work-life balance by disconnecting from work communication after work hours."
```

---

**FARAD Group #0000 Article 1 (Author GPT4o)**

In the bustling city of Fuzhou, where tradition meets the modern hustle, Zhao Wei stands as a beacon of innovation and resilience. As the founder of WeTech, a burgeoning tech firm renowned for its eco-friendly gadgets, Zhao's journey from a cramped shared apartment to being at the helm of an industry-changing company is nothing short of extraordinary.

### The Humble Beginnings of WeTech

Zhao Wei's story begins with the simplest of settings--a shared apartment with only the most basic resources at his disposal. "I remember those early days vividly", Zhao remarks in an interview. "Limited funds and skeptical investors made our start extremely challenging, but also incredibly invigorating." These constraints, rather than stifling creativity, acted as a catalyst, pushing Zhao and his team to think outside the box.

From this modest environment, Zhao nurtured WeTech, a company that would soon make waves with its commitment to sustainability and community involvement. Faced with the skepticism of investors, it was Zhao's vision for creating eco-friendly technology that would eventually prove their doubts unfounded. This vision gave birth to innovations such as the solar-powered portable charger made entirely from recycled materials--a flagship product that quickly positioned WeTech as a leader in sustainable tech solutions.

### A Culture of Empowerment

A significant aspect of WeTech's success can be attributed to the distinctive work culture Zhao Wei has cultivated. Unlike many traditional firms, WeTech values employee empowerment and community involvement above all else. Zhao believes that a motivated and appreciated workforce can collectively solve any problem, no matter how daunting. "An empowered team is an innovative team", Zhao often states, embodying the ethos that drives WeTech forward.

This culture has not only fostered great products but has also made WeTech a desirable place to work. Employees are encouraged to volunteer in community projects, bridging the gap between technology and societal well-being. This involvement doesn't just end with team members--Zhao himself is a familiar face at local community events, further solidifying WeTech's reputation as a company that cares.

### Mentoring the Next Generation

Beyond his organizational achievements, Zhao Wei has taken it upon himself to mentor the next generation of entrepreneurs. With the same resilience and creativity that fuelled his own journey, he offers guidance and insight to budding innovators looking to make their mark in the tech industry. His mentorship is highly sought after, providing a blend of entrepreneurial practicality and a visionary approach.

Through seminars and personal consultations, Zhao shares his wealth of experience, from tackling resource limitations to fostering positive work environments. "Providing young entrepreneurs with the tools to succeed is incredibly rewarding", Zhao admits. "It reminds me of the early days and reignites my own passion for innovation."

### Balancing the Professional with the Personal

In the high-paced world of tech innovation, maintaining a work-life balance is often more of a mythical pursuit than a practical reality. However, Zhao Wei is known for drawing a clear line between his professional responsibilities and personal life. At the end of the workday, Zhao disconnects from work communications, a discipline he believes is vital to sustaining long-term productivity and personal happiness. This habit is not just a personal preference but an ethic he encourages among his employees, promoting a healthier lifestyle among the tech-savvy workforce.

### An Exemplar of Resilience and Creativity

Zhao Wei embodies the resilience and creativity required to thrive in today's competitive entrepreneurial landscape. Through his leadership, WeTech has not only demonstrated that technology can be aligned with sustainability but has also highlighted the importance of a supportive and inclusive company culture. Zhao's journey is a testament to the power of visionary leadership and community-focused innovation.

In Zhao's own words, his journey with WeTech is far from over. "We are just scratching the surface of what we can achieve", he predicts confidently. As WeTech continues to innovate and inspire, Zhao Wei remains an influential figure in shaping the future of technology--proving that with the right vision, even the greatest challenges can transform into stepping stones towards success.

## F.4 EXAMPLE WATERMARKED DOCUMENT AND CORRESPONDING QUESTIONS

---

**FARAD Group #0000 Article 1 (Author GPT4o): Watermarked Version by WARD**

In the vibrant metropolis of Fuzhou, where age-old traditions blend seamlessly with the relentless pace of modern life, Zhao Wei stands out as a shining embodiment of entrepreneurial spirit and determination. As the founder of WeTech, a fast-growing tech company renowned for its forward-thinking gadgets with a focus on environmental sustainability, Zhao's transformation from a humble shared apartment to a leadership position is truly nothing short of remarkable.

### Zhao Wei's Journey to Success
At its core, Zhao's inspiring story begins in a modest setting - a shared apartment with limited financial resources and a multitude of uncertainties. "Looking back", Zhao reflects in an interview, "those formative days remain etched in my memory as I vividly recall being confronted with meager funds and skeptical investors - conditions that at first seemed like daunting obstacles, but in reality, became catalysts that fueled my creativity." Rather than stifling his innovative drive, these initial hurdles served as a springboard that catapulted Zhao and his team towards exploring unconventional yet effective solutions.

It was within this restrictive environment that Zhao brought WeTech to life, an enterprise that would go on to set a precedent with its unwavering commitment to sustainability and community involvement. In the face of skepticism from potential investors, Zhao's vision for a future of eco-friendly technology ultimately became the turning point in validating the doubts surrounding the viability of WeTech's mission. Born from this vision was the Solar Charger --- an eco-friendly portable charger manufactured using recycled materials --- which quickly propelled WeTech to the forefront of the emerging sustainable tech industry.

### Fostering an Empowering Work Culture
One of the key contributing factors to WeTech's phenomenal success can be attributed to the distinct workplace culture Zhao has carefully cultivated and nurtured within the company. Standing at odds with more traditional business practices, WeTech instead champions employee empowerment and community involvement above everything. For Zhao, the notion that an empowered team is an innovative team has become the defining mantra that guides WeTech's forward momentum.

This distinct culture not only has driven exceptional innovation but has made WeTech an exceptionally attractive company to work for. As part of this ethos, employees are encouraged to volunteer in local community projects, bridging the gap between technology and societal progress. Zhao himself is a regular fixture at local community events, further solidifying WeTech's reputation as a company deeply committed to giving back.

### Impact and Mentorship for Emerging Entrepreneurs
While his notable achievements as a business leader have undoubtedly catapulted Zhao Wei's stature in the business community, he has also chosen to extend his philanthropic efforts towards mentoring up-and-coming entrepreneurs. Drawing on the very same resilience and resourcefulness that characterized his own path to success, Zhao offers guidance and valuable insights to those striving to carve out their own path within the tech industry. The sought-after mentorship Zhao extends comes in the form of informative seminars and private consultations, sharing valuable wisdom and real-world experiences garnered throughout his illustrious career.

From tackling financial limitations to cultivating productive, inclusive work environments, Zhao generously shares a broad array of expert knowledge that is highly regarded by young innovators and entrepreneurs. "Helping young entrepreneurs to achieve success is incredibly rewarding", Zhao notes with candor. "It also reignites within me my boundless passion for innovation."

### Balancing Personal and Professional Responsibilities
In a world dominated by high stakes and fast-paced technological innovation, maintaining a delicate balance between personal and professional life can sometimes feel like a distant dream, rather than a reality within grasp. Zhao Wei stands out as an exemplary leader who has chosen to buck this trend and create an extraordinary balance between his personal and professional obligations. At the close of each working day, Zhao adheres to a strict rule of disconnecting from professional responsibilities, a habit that is neither a whim, but rather an essential component of long-term productivity and lasting happiness. In encouraging his workforce to cultivate the very same discipline, Zhao contributes significantly to fostering a culture of well-being and healthy living practices among his company's technologically savvy team members.

### Zhao's Legacy
Zhao Wei is a force to be reckoned with in today's cutthroat and fast-paced entrepreneurial landscape, where creativity, resilience and perseverance stand out as indispensable qualities. Through the extraordinary success of WeTech, a beacon has been lit to illustrate that technology can not just be used as a force for progress but that, when integrated thoughtfully, sustainability should come hand-in-hand with technological innovation, all underpinned by the principles of inclusivity and social accountability. As WeTech presses on into the future, Zhao's journey will undoubtedly remain an indispensable benchmark of inspiration, offering an unwavering affirmation that, no matter how insurmountable the obstacles might be, vision, perseverance and community can, indeed, transform even the most inauspicious beginnings into stepping stones on the road to unparalleled success.

---

**Question Generated by WARD to use as a Query to the RAG System**

How does Zhao Wei's personal approach to business and leadership--emphasizing sustainability, community involvement, employee empowerment, mentorship, and personal balance--contribute to the unique identity and success of WeTech in the competitive tech industry, and what implications does it have for emerging entrepreneurs who are inspired by his journey?

## F.5  PROMPTS USED BY WARD

**Watermarked Paraphraser Prompt**

You are an expert rewriter. Rewrite the following document keeping its meaning and fluency and especially length. It is crucial to retain all factual information in the original document. DO NOT MAKE THE TEXT SHORTER. Do not start your response by 'Sure' or anything similar, simply output the paraphrased document directly. Do not add stylistic elements or anything similar, try to be faithful to the original content and style of writing. Do not be too formal. Keep all the factual information.

**Question Generation Prompt**

Given a document, generate a question that can only be answered by reading the document. The answer should be a longer detailed response, so avoid factual and simple yes/no questions and steer more towards questions that ask for opinions or explanations of events or topics described in the documents. Do not provide the answer, provide just the question.

**Question Generation Prompt for $qpd > 1$ (multiple questions per document)**

Given a document, generate exactly $qpd$ questions that can only be answered by reading the document. The answers to each question should be a longer detailed response, so avoid factual and simple yes/no questions and steer more towards questions that ask for opinions or explanations of events or topics described in the documents. Do not provide the answers, provide just the questions. Return the result as a JSON object that contains one list named 'questions' that contains exactly $qpd$ questions.

## F.6  PROMPTS USED BY THE BASELINES

**FACTS Question Generation Prompt**

For a given document, generate a single simple question that can only be answered by reading the document, and is absolutely unanswerable for someone who has not read the document.

**FACTS Query Template**

<Question>
Do not hallucinate the answer. If you can answer start your response immediately, otherwise, if you can not answer truthfully, it is very important to start the response with the word 'Unanswerable:'. Feel free to explain why it is unanswerable after, but always start such responses with the word Unanswerable.

**AAG Query Template**

Does this:
'<Document>'
appear in the context? Answer with Yes or No.

**SIB Query Template**

Continue this, writing a longer response: <Prefix of the Document>

## F.7 PROMPTS USED FOR LLM TEXT QUALITY JUDGING

---

**Judging (Watermarked) Documents**

```
Given a document, grade its quality and text fluency as an INTEGER on a scale of 1 to 10. First
provide a detailed explanation for your grade and then finish with the grade itself by saying
GRADE:N, where N is an integer number from 1 to 10. Do not finish the response with a period;
finish exactly with the number.
```

---

**Judging RAG Responses with/without Watermarked Data**

```
You are given an interaction of a user with a question answering system. ###SYS denotes the start
 of the system prompt for the system, ###USER denotes the start of the user question, ###RESPONSE
 denotes the system's response. Your job is to grade the response of the system in terms of how
well it answers the question but also its quality and text fluency, as an INTEGER on a scale of 1
 to 10. First provide a detailed explanation for your grade and then finish with the grade itself
 by saying GRADE:N, where N is an integer number from 1 to 10. Do not finish the response with a
period; finish exactly with the number.
```

