# OpenReview forum: "Ward: Provable RAG Dataset Inference via LLM Watermarks"
_ICLR.cc/2025/Conference — ICLR 2025 Poster_

### Official Review · Reviewer_naE4 · 2024-10-29

**Soundness:** 3
**Presentation:** 3
**Contribution:** 3
**Rating:** 6
**Confidence:** 3

**Summary:**

This paper tackles the issue of unauthorized data use in RAG systems. Noting the current absence of tools for detecting such misuse, the authors propose "RAG Dataset Inference" (RAG-DI), a framework that empowers data owners to identify unauthorized data utilization in RAG systems. To facilitate research on RAG-DI, they developed FARAD, a synthetic dataset generated by leveraging an existing dataset and LLMs. FARAD is crafted to simulate challenging scenarios where similar content may appear across multiple sources, enhancing its applicability in real-world settings.

Furthermore, the authors introduce WARD, a novel method that leverages an existing watermarking algorithm to enable data owners to statistically verify whether their data has been incorporated into an RAG system's corpus through black-box queries. Experimental results demonstrate that WARD consistently outperforms existing methods, providing remarkable accuracy, robustness, and efficiency. This approach offers a reliable mechanism for detecting unauthorized data use and establishes essential resources and methodologies to guide future research on data privacy in RAG systems.

**Strengths:**

1. Although RAG has gained popularity, its implications for data privacy and unauthorized usage detection remain largely unexamined. This paper introduces the RAG Dataset Inference (RAG-DI) concept alongside a watermark-based detection approach, offering a novel perspective on privacy within RAG frameworks. By developing the FARAD dataset and the watermark-based WARD method, this work establishes a strong foundation for future research. It stands as a pioneering contribution to addressing privacy concerns in RAG systems.

2. The paper offers comprehensive experimentation and empirical evaluation to validate WARD. The paper demonstrates WARD's outstanding performance across challenging scenarios by benchmarking WARD against multiple baselines, including adaptations from related fields.

3. The paper is well-written and organized, with a clear structure that enhances readability. The authors present their work logically, making the flow of ideas easy to follow in each section.

**Weaknesses:**

1. The authors claim their approach is effective in the challenging scenario where information is distributed across multiple sources. However, it remains unclear how the method performs when only one retrieved document is watermarked while others are unprotected. Additionally, they evaluate the method only with k=3; it is uncertain how effective it would be if the system retrieves more, say 10 relevant documents, with only one being protected. Without this analysis, the effectiveness of the proposed approach cannot be conclusively assessed, especially given that RAG systems can flexibly retrieve a variable number of relevant documents. Moreover, the "provable watermark" claim lacks empirical support; no rigorous evidence is provided to substantiate this assertion. While “provable” may reference the watermarking algorithm itself, scalability should also be addressed within this definition to strengthen the claim.

2. The authors claim their approach is provable; however, unlike existing watermarking algorithms, they do not provide a rigorous proof. This discrepancy undermines the study’s credibility.

3. The authors propose a relatively simple defense strategy that relies solely on prompt engineering. However, due to the complex requirements of the proposed defense, LLMs may struggle to adhere to instructions precisely, resulting in a suboptimal defense. A more practical and effective approach could involve monitoring the words used in the retrieved documents and limiting or penalizing the frequency of these words in generated responses. Additionally, after the response is generated, it could be further paraphrased to potentially reduce the usage of watermarking words, similar to techniques employed in existing watermarking approaches. Again, the issue is rooted in the absence of a theoretical guarantee.

**Questions:**

1. To my understanding, generating watermarked text typically requires white-box access to LLMs to direct the generation process. Then, how can one produce watermarked documents using black-box models like GPT-4 and Claude 3.5-Sonnet? Could you elaborate on the methods or approaches that make this possible?

---

> ### Author Response · Authors · 2024-11-21
> **Response to Reviewer naE4 (Comment 1/2)**
>
> We thank the reviewer for their detailed feedback. We reply to the reviewer’s concerns below, numbered Q1-Q6, and point out that we uploaded a new revision of the paper, with new additions marked in purple. We are happy to discuss any outstanding points further.
>
> **Q1: How does Ward perform when only one of the retrieved documents is watermarked?**\
> As this in fact holds in all our experimental runs in the _IN_ case, we believe there may be a misunderstanding here, possibly caused by our misleading formulation of the perfect retrieval mechanism on L340. In our new revision we replace that wording and point to the new App. E, which contains a precise explanation of perfect retrieval along with two examples. In short, (1) each query of Ward is done using a single document, thus we do not expect more than one watermarked document to end up in the context; and (2) as we set $k \geq 3$ in our experiments, the context will contain “distraction” documents, which in the Hard setting will primarily be from the same FARAD group, but may also be from other groups. We hope this resolves the question, and we encourage the reviewer to follow up otherwise.
>
> **Q2: Does Ward still perform well with more than 3 retrieved documents in RAG?**\
> Yes—we point the reviewer to our experiments with $k=4$ and $k=5$ presented in App. A.1 and referenced on L485, that the reviewer may have missed. Our choices here were guided by prior work on RAG such as e.g., [“FlashRAG: A Modular Toolkit for Efficient Retrieval-Augmented Generation Research”, Jin et al. 2024] which recommends $k \in [3,5]$ (Figure 2, Left). This matches the values used in the RAG MIA baselines introduced in Sec. 2 [Li et al., 2024, Anderson et al., 2024], which set k=5 and k=4 respectively.
>
> Nonetheless, we follow the reviewers suggestion and run a new experiment with Ward with $k=10$, FARAD-Hard, both Naive-P and Def-P prompts, 5 seeds, and Claude 3 Haiku (as this experiment often exceeds the context size of GPT-3.5, which was used in our experiments in App. A.1). All 10 runs were successful in both _IN_ and _OUT_ cases, and Ward converged to 100% accuracy in around 50 queries, similar to baseline Haiku results in Fig. 5 (Left). This reaffirms our conclusion that Ward is robust to different values of $k$. We will extend this experiment further and add the complete results to the next version of App. A.1.
>
> **Q3: Can you empirically support the provability claims made in the paper, and provide corresponding rigorous proofs?**\
> We would like to first clarify that the notion of “provability” that we consider, and claim holds for Ward, is only the one we introduce on L241 under _Guarantees_, i.e., well-controlled small rate of False Positives / type 1 errors. In other words, if Ward concludes that the data was used in a certain RAG system, this claim is false only with a very small known probability. This is the standard notion of provability in LLM watermarking, naturally motivated by the high cost of false positives, which is certainly the case in RAG-DI. In fact, as noted on L397, our proof would also directly follow from the proof for the underlying scheme of Ward [Kirchenbauer et al., 2024], thus we did not repeat it. We agree however that formalizing this within our paper could be beneficial and make the paper more self-sufficient, so we are happy to work on this for the next paper revision. We also point out Fig. 6 and Fig. 8 as relevant, that empirically show the validity of our p-values, which quickly drop as we increase the number of queries for _IN_ cases (true positives), but stay consistently close to the expected value of 0.5 for _OUT_ cases (no false positives).
>
> We certainly had no intention to claim that Ward is _provably_ always able to discover unauthorized data usage, i.e., that it has a perfect rate of True Positives, as this is in principle impossible. For example, if an LLM refuses every request, the worlds where our data was (_IN_) or was not used (_OUT_) are indistinguishable. Thus, we note that aspects like scalability and robustness are strictly orthogonal to the discussion around provability. Of course, they are still crucial as empirical evidence of the efficacy of Ward, and we have investigated them in detail in our experiments in Sec. 5, showing that Ward is efficient and robust to many setting variants. Regarding scalability in particular, we have now included an extensive discussion in our new App. F on practicality and future work (paragraph “Efficiency”).
>
> We are happy to consider any concrete suggestions that the reviewer has around how to improve the phrasing of provability in the paper.

---

> > ### Author Response · Authors · 2024-11-21
> > **Response to Reviewer naE4 (Comment 2/2)**
> >
> > **Q4: Can you evaluate a defense that penalizes response tokens based on their presence in the retrieved documents?**\
> > We agree with the reviewer that prompting defenses have limitations, which originally motivated us to include an experiment where the RAG provider uses a more powerful defense based on MemFree decoding [Ippolito et al., 2023], that encapsulates the idea proposed by the reviewer. This defense aims to reduce the propagation of the watermarking signal by preventing n-gram overlaps between the response and the retrieved documents. As discussed in L422-425, Ward maintains full accuracy in this setting. We hope this answers the reviewer’s question.
> >
> > **Q5: Can you discuss other possible defenses, such as paraphrasing of responses?**\
> > Certainly. We have now added a detailed discussion of countermeasures to our new App. F that discusses practical considerations and future work, and referenced it in the main paper. We discuss ideas such as preprocessing of the RAG corpus or postprocessing of the LLM responses to remove the watermark. We argue that as identifying watermarked documents is hard, the RAG provider would need to paraphrase _every_ incoming article, which incurs a significant overhead and is not practically feasible. We conclude that from this perspective, it may be more feasible for the RAG provider to legally acquire the data, which is an interesting positive side-effect of having practical RAG-DI tools. Of course, future work may develop more sophisticated defenses against Ward, which we certainly welcome.
> >
> > **Q6: How do you watermark documents without white-box access to an LLM?**\
> > We apologize for the confusion—as the reviewer finds more natural, we assume that the data owner has white-box access to an OSS LLM. As this information was missing on top of Sec. 5, we have introduced it (L343) in our latest revision. Independently, we point out that black-box watermarking is an interesting research question in itself, and we are aware of several works that try to achieve this via rewriting [1,2] or rejection sampling [3].
> >
> > [1] “Watermarking Text Generated by Black-Box Language Models”, Yang et al., 2023 \
> > [2] “PostMark: A Robust Blackbox Watermark for Large Language Models”, Chang et al., 2024 \
> > [3] “A Watermark for Black-box Language Models”, Bahri et al., 2024

---

> > > ### Comment · Reviewer_naE4 · 2024-11-27
> > >
> > > Thanks for the detailed feedback. Most of the concerns have been addressed, but I have one left:
> > >
> > > Given that your watermarking method relies on the green-red list, which requires white-box access, how do you obtain such access to commercial models to watermark the synthetic documents?

---

> > > > ### Author Response · Authors · 2024-11-27
> > > >
> > > > $\newcommand{mam}{\mathcal{M}}$We thank the reviewer for following up; we are pleased to hear that most concerns have been resolved.
> > > >
> > > > To clarify the remaining point further, we remark that there are two separate LMs under consideration here: (1) the model we are auditing ($\mam$), deployed by the RAG provider as part of a RAG system; and (2) the auxiliary model used by the data owner to watermark $D_{\text{do}}$ before applying Ward (let’s denote this LM with $\mam_{\text{do}}$).
> > > >
> > > > We note that the data owner has full control of the watermarking step, so they are able to use any open-source model as $\mam_{\text{do}}$, which enables the use of red-green watermarks, as the reviewer correctly points out. Concretely, in our experiments in Sec. 5 we set $\mam_{\text{do}}=\text{{Llama3.1-8B}}$ (see L343).
> > > >
> > > > Importantly, we do not make any white-box assumptions on $\mathcal{M}$, the model that is audited by using Ward—this enables us to use popular closed-source API models like $\text{{Claude 3 Haiku}}$ and $\text{{GPT-3.5}}$ as $\mam$ in our evaluation in Sec. 5.

---

> > > > > ### Author Response · Authors · 2024-12-01
> > > > >
> > > > > We thank the reviewer again for engaging in the discussion. Does our last comment address the remaining concern? If so, we would appreciate if the reviewer could consider increasing their score to reflect that.

---

### Official Review · Reviewer_nePY · 2024-11-04

**Soundness:** 3
**Presentation:** 3
**Contribution:** 3
**Rating:** 6
**Confidence:** 3

**Summary:**

This paper focuses on the problem of unauthorized use of datasets in Retrieval-Augmented Generation (RAG) systems and formalizes it as the RAG Dataset Inference (RAG - DI) problem. To address this issue, the authors propose the FARAD dataset specifically designed for RAG - DI evaluation and a series of baseline methods. They also introduce the WARD method based on LLM watermarks. Experiments demonstrate that WARD outperforms the baseline methods in terms of accuracy, query efficiency, and robustness.

**Strengths:**

1. The RAG - DI problem is formally defined, filling the research gap in this field and laying a foundation for subsequent research.
2.The design of the FARAD dataset takes into account the practical application scenarios of the RAG system and avoids the shortcomings of existing datasets in RAG - DI research, such as the possibility of being used for LLM training and the lack of fact redundancy.
3.The WARD method is based on LLM watermarks and can provide data owners with strict statistical guarantees regarding the use of their datasets in the RAG system. It exhibits excellent performance in various experimental settings and outperforms all baseline methods.

**Weaknesses:**

1. In real-world applications, RAG systems may face more complex situations, such as multilingual environments and data from different domains. The discussion in this regard in the paper is relatively limited.
2. There is a lack of discussion on the efficiency of the solution.

**Questions:**

1. For different types of LLMs, will the performance of the WARD method be affected? If so, how to adjust and optimize it?
2. In practical applications, the update frequency of data may be very high. What impact will this have on the effectiveness of the WARD method? How to deal with this situation?

---

> ### Author Response · Authors · 2024-11-21
> **Response to Reviewer nePY**
>
> We thank the reviewer for their valuable feedback. We respond to the reviewer’s concerns below, numbered Q1-Q4, and point out that we uploaded a new revision of the paper, with new additions marked in purple. We are happy to continue the discussion in case of follow-up questions.
>
> **Q1: Can you discuss how the data used in your work models complex real-world scenarios?**\
> From the model owner’s side, while we certainly agree that this can be further improved, we believe FARAD is a solid baseline for future work in terms of how well it reflects realistic cases, greatly improving over the prior state where there were no suitable datasets that (1) are provably not leaked and (2) contain fact redundancy. In a newly added App. F (referenced from Sec 3.1), we recap the key design decisions of FARAD, and explicitly point out several promising directions in which FARAD can be improved in future work, including the directions suggested by the reviewer (domain and language shift), which would enable the evaluation of Ward in more diverse settings. From the RAG provider’s side, we remark that the presence of documents in other languages or from other domains in the RAG corpus should generally not affect the performance of Ward as a RAG-DI method. In particular, if the RAG pipeline effectively handles the challenge of a multilingual and/or multidomain corpus, Ward should be directly applicable.
>
> **Q2: Can you discuss the efficiency of Ward?**\
> Certainly. While our first draft contained some efficiency/complexity discussion, evaluating Ward in terms of query cost, and showing that at most 100 queries are needed for a confident decision across all settings, we acknowledge that this concern was grouped with Monotonicity and not extensively discussed in one place. In the new revision we added a unified discussion of Efficiency to our new App. F focused on practicality and future work, and referenced it in the main paper.
>
> In particular, we translate our query cost result above into dollar cost of closed-model APIs, concluding that only \\$0.63 is needed to apply Ward to GPT-3.5, and at most \\$2 for any popular state-of-the-art closed model. We further discuss why the query complexity of Ward is independent of the size of both the RAG corpus $D$ and the owner dataset $D_{do}$, substantiate that large $|D_{do}|$ is not an obstacle. All these results point at Ward being practically applicable. Finally, in App. F we point out our preliminary attempts to further improve the sample efficiency of Ward (App. A.4), and conclude with ideas for future work.
>
> **Q3: How is the performance of Ward affected by the type of LLM used?**\
> Our main experiments, performed on the Llama3.1-70B open model and two popular closed models, have demonstrated that Ward is robust to this change. We do however observe the difference in the “difficulty” of the task for different models, which we hypothesize is often due to the models’ approach to following instructions such as the defense prompt. We certainly find it interesting to further quantify this, but remark that we foresee no case where Ward would fail due to the choice of the LLM, especially as LLM watermarks were demonstrated to work reasonably well on a wide range of models [Piet et al. 2023, Kirchenbauer et al. 2024], and most recently on the production deployment of Gemini [“Scalable watermarking for identifying large
> language model outputs”, Datathri et al., 2024, Nature]. We are happy to discuss this further if the reviewer has follow-up questions.
>
> **Q4: How does the update frequency of data impact Ward?**\
> Due to the fundamental properties of RAG, we do not believe this significantly impacts the success of Ward. As long as the data used for querying was present in the RAG corpus throughout the querying, and the retrieval mechanism works roughly as expected, Ward should generally be successful, as demonstrated in our experiments. This holds independently of other documents being added/removed, and such changes to the RAG corpus do not necessitate any modifications to Ward before it is run.

---

> > ### Author Response · Authors · 2024-12-01
> >
> > We would like to respectfully follow up with the reviewer and remind them to acknowledge our rebuttal above. Does the reviewer find that we have addressed all their concerns? If so, we would appreciate if they could consider increasing their score to reflect that.

---

### Official Review · Reviewer_WHEe · 2024-11-05

**Soundness:** 3
**Presentation:** 3
**Contribution:** 2
**Rating:** 6
**Confidence:** 4

**Summary:**

This paper presents WARD, a novel method for detecting unauthorized dataset usage in Retrieval-Augmented Generation (RAG) systems. The approach formalizes RAG Dataset Inference (RAG-DI) as a problem where data owners seek to identify if their data is being used without permission in RAG systems via black-box queries. The authors introduce a new dataset, FARAD, specifically designed to benchmark RAG-DI under realistic conditions that include fact redundancy. Using LLM watermarks, WARD offers data owners statistical guarantees of usage detection. Experimental results show that WARD outperforms other baselines, demonstrating high accuracy, query efficiency, and robustness across challenging scenarios.

**Strengths:**

The strengths of this paper are mainly centered around the propose of the new task along with a new specialized benchmark dataset.

1. Novel Problem Definition and Formalization: The paper identifies and formalizes the novel problem of RAG Dataset Inference (RAG-DI), addressing a critical need for data owners to detect unauthorized data usage in RAG systems. This formalization fills a significant research gap and sets the stage for further exploration of secure data usage in RAG contexts.

2. Introduction of a Specialized Benchmark Dataset: By creating FARAD, the authors contribute a valuable dataset that is specifically tailored for RAG-DI research. FARAD’s design ensures realistic conditions for testing, including the incorporation of fact redundancy, making it highly relevant for practical evaluations of RAG-DI approaches.

3. Comprehensive Experimental Validation: The paper’s extensive experiments demonstrate the effectiveness of WARD in achieving high true positive rates with no false positives across various settings, underscoring the method's robustness and accuracy. This comprehensive validation strengthens the credibility of WARD as a reliable solution for detecting unauthorized data use in RAG systems.

**Weaknesses:**

Although this paper has the above strengths that are interesting, I also noticed several weaknesses that should be further considered.

1. Application Scenario. The authors propose a novel problem definition, namely RAG Dataset Inference (RAG-DI). The concept is quite straightforward, and one has to admit that the concern indeed exists in practice. However, it is still less discussed in this paper, that how often this type of problem can occur in realistic scenarios. I am concerned with the application range of this work. For example, if a person is concerned with their own data being used for LLMs, do they need to test all LLMs in the market? This could be prohibitive.

2. Lack of Computation Complexity. The authors proposed a novel method to deal with the specific problem of RAG-DI. However, the complexity is not fully discussed in the paper. As a result, the problem of scalability could be severe. For example, the owner has an excessively large dataset, will the method still work?

3. Countermeasures: Although the authors considered the scenario where the RAG provider attempts to prevent the unintended use of the system, they do not provide a detailed discussion of the possible countermeasures that the RAG provide may use to counteract the watermarking of data. As there are multiple existing techniques against watermarking, the discussion about vulnerability of the proposed framework when faced with these countermeasures could be inspiring

**Questions:**

How can this framework be applied to realistic scenarios?

---

> ### Author Response · Authors · 2024-11-21
> **Response to Reviewer WHEe**
>
> We thank the reviewer for their detailed comments. We respond to all questions raised by the reviewer below, numbered Q1-Q4, and point out that we uploaded a new revision of the paper, with new additions marked in purple. We are happy to continue the discussion further if the reviewer has follow-up questions.
>
> **Q1: Can you discuss the importance of RAG-DI and the applicability of Ward in realistic scenarios?**\
> We are glad that the reviewer agrees that the concern underlying RAG-DI exists in practice, and we acknowledge that the paper could have included a more thorough discussion of the motivation and the importance of this problem setting, example use-cases, and the practical applicability of Ward. We now introduce all of these in a new App. D, referenced in the first paragraph of Sec. 1. The applicability of Ward is also extensively discussed in our new App. F from the perspective of efficiency (see Q3 below) and countermeasures (see Q4 below). We encourage the reviewer to point out if there are aspects they believe we missed in this new discussion, and we are happy to extend this section accordingly.
>
> **Q2: Does the data owner need to query every suspect LLM to test for unauthorized data usage?**\
> As we discuss in the new App. D, this is naturally true, as querying $\mathcal{M}$ is necessary to make any claim about $\mathcal{M}$. However, we do not see this as prohibitive, for several reasons. First, a single application of Ward is very cheap, below \\$2 for the whole process of RAG-DI, with potential for further improvement (see Q3 below). Second, the ecosystem of relevant LLM providers is relatively small. In particular, not many providers both (i) have resources for indiscriminate large-scale scraping of data and (ii) are sufficiently popular to create market harm based on the unauthorized usage of that data. Thus, we believe applying Ward in real use-cases is practical from an efficiency and cost standpoint.
>
> **Q3: Is scalability an issue for Ward? What if $D_{do}$ is large?**\
> Good question. While our first draft contained some efficiency/complexity discussion, evaluating Ward in terms of query cost, and showing that at most 100 queries are needed for a confident decision across all settings, we acknowledge that this concern was grouped with Monotonicity and not extensively discussed in a single place. In the new revision we added a unified discussion of Efficiency to our new App. F focused on practicality and future work, and referenced it in the main paper.
>
> In particular, we translate our query cost result above into dollar cost of closed-model APIs, concluding that only \\$0.63 is needed to apply Ward to GPT-3.5, and at most \\$2 for any popular state-of-the-art closed model. We further discuss why the query complexity of Ward is independent of the size of both the RAG corpus $D$ and the owner dataset $D_{do}$, and substantiate that large $|D_{do}|$ is not an obstacle, as the data owner by design uses only a subset of those as queries, and can also decide to not watermark all of $D_{do}$. All these results substantiate our argument that Ward is viable to apply in practice. Finally, in App. F we now point out our preliminary attempts to further improve the sample efficiency of Ward (App. A.4), and conclude with ideas for future work in this direction, such as using a preliminary round of a few queries to establish _reasonable suspicion_, followed by the full application of Ward (while making sure to preserve the statistical soundness).
>
> **Q4: Can you discuss the possible countermeasures to Ward?**\
> Yes—similar to Q3, we have added a detailed discussion of countermeasures to our new App. F and referenced it in the main paper. To summarize these new additions, we first point out that App. A.5 demonstrates the robustness of Ward against a tailored defense based on MemFree decoding [Ippolito et al., 2023] which prevents n-gram overlaps between the response and the retrieved documents. We proceed to discuss ideas such as preprocessing of the RAG corpus or postprocessing of the LLM responses to remove the watermark. We argue that as identifying watermarked documents is hard, the RAG provider would need to paraphrase _every_ incoming article, which incurs a significant overhead and is not practically feasible. We conclude that from this perspective, it may be more feasible for the RAG provider to legally acquire the data, which is an interesting positive side-effect of having practical RAG-DI tools. Of course, future work may develop more sophisticated defenses against Ward, which we certainly welcome.

---

> ### Comment · Reviewer_WHEe · 2024-11-26
>
> Thanks for your detailed concerns. I think my concerns have been thoroughly addressed. I will raise the score to 6.

---

> > ### Author Response · Authors · 2024-12-01
> >
> > We thank the reviewer for acknowledging our rebuttal, and rating the paper as marginally above acceptance. We are glad that the reviewer explicitly states that all their concerns were thoroughly addressed; could they consider adjusting their score further to reflect this? Otherwise, if there are in fact additional concerns that prevent this, we are happy to hear them and use the remaining time to address them.

---

### Official Review · Reviewer_Lfcg · 2024-11-11

**Soundness:** 3
**Presentation:** 4
**Contribution:** 3
**Rating:** 8
**Confidence:** 3

**Summary:**

This paper studies the problem of RAG Dataset Inference (RAG-DI), where a data owner aims to detect whether their dataset is included in a RAG corpus via black-box queries. The key difference from the RAG MIAs setup is that the data owner makes a dataset-level decision, instead of a document-level decision as in MIAs.
The authors first introduce a dataset FARAD for the RAG-DI problem, as they believe the previous datasets used in MIAs problems (EnronEmails, HealthcareMagic) are likely contaminated in the training datasets of existing LLMs and lack fact redundancy. The FARAD dataset builds on a recent source of articles with fictional entities and events, and they prompt a set of LLMs (author LLMs) to rewrite articles based on the key points from the source article.

The paper then adapts existing RAG MIAs baselines to this problem and proposes a simple baseline called FACTS that simply prompts an LLM to generate a single question that is only answerable by the document. They find that the FACTS baseline performs well in their easy setting (without fact redundancy) but not in their hard setting (with fact redundancy).

Finally, they propose a new dataset inference method based on LLM watermarking called Ward. The key idea is to paraphrase the articles with a watermarked model from the recent LLM watermarking literature. They show that Ward works well in their hard setting and also meets the desiderata including monotonicity, guarantees, and robustness.

**Strengths:**

* I believe the paper studies an important and timely research question that has been overlooked in the literature. I am not a domain expert (in the sense of MIA and RAG privacy), so I may not know related work very well.
* I think the paper has made many meaningful contributions, from formulating the problem to collecting the dataset and adapting RAG-MIA baselines, to finally proposing their own approach based on LLM watermarking. The experiments (different LLMs and evaluation settings)  and analyses (hyperparameters, modeling retrieval, and examining the quality of paraphrased texts) are quite thorough too.
* Using LLM watermarking as an approach to solve dataset inference seems a novel and reasonable approach to me.
* The paper is very well written and self-contained. It is easy to follow the structure of the paper and understand its contributions clearly. I have learned a lot from reading the paper.

**Weaknesses:**

The major part I am not very sure about is whether the dataset construction and experimental settings can really reflect realistic RAG settings in real-world deployment, since the effectiveness of the baselines and proposed approach all depends on this setting. In particular:
* It assumes perfect retrieval (= only articles from the same source are used) in most experiments. Although the paper discusses a more practical retrieval setting in Section 5.3, I don't see any discussion of previous baselines. For the hard setting, the imperfect retrieval problem should be easier than the perfect retrieval settings (the distracting documents should be less relevant), so the baselines could probably perform better.
* I am not entirely sure whether the current data generation pipeline can well reflect the redundancy problem in the real world, given that they are all generated from different LLMs (I am also not sure how much impact "additional facts" and "a set of diverse LLMs" have here). It might be too much to ask, but I think having a realistic multi-document summarization setting would be more realistic.
* I have difficulty understanding the rationale behind the defense prompt. Why would RAG providers want to instruct the models to not answer questions from retrieved documents? How can RAG be effective in this case? If this is from the data owner's perspective, given the goal is to detect whether the dataset is used or not, I don't see why they would want to add such a prompt to make the detection task harder.
* [Question] The RAG uses k = 3 shots, but the Easy setting assumes there is only one article per group. Where do the 3 shots come from? The paper also mentions they ablate the choice of k, where k can be {3, 4, 5} in the Appendix. Even for the Hard setting, aren't there only 4 articles per group? Not sure if I misunderstood this.

I still think it is a good paper regardless, and I hope to get some clarifications on these questions. There is a chance that I just misunderstood some key parts of the paper.

[Minor] It is not necessarily a weakness, but the paper's title is "Ward:... Watermarks" while the paper really contains much more than that, including setting up the problem, establishing the dataset and baselines. It would probably be more appropriate to have a broader title around RAG Dataset Inference.

**Questions:**

See weaknesses.

---

> ### Author Response · Authors · 2024-11-21
> **Response to Reviewer Lfcg**
>
> We thank the reviewer for their comprehensive feedback. We respond to the raised concerns below, numbered Q1-Q4, and point out that we uploaded a new revision of the paper, with new additions marked in purple. We are happy to continue the discussion further if the reviewer has additional questions.
>
> We appreciate the title change suggestion and the author’s acknowledgement of the breadth of our work, and are discussing several options among the coauthors.
>
> **Q1: How does perfect retrieval work for k=3 (Easy) and k=5 (Hard)? Does it always retrieve only articles from the same source/group?**\
> No; this is a misunderstanding caused by our misleading wording on L340. In our new revision we remove that wording and point to the new App. E, which now contains a precise explanation of perfect retrieval along with two examples that exactly correspond to the two cases outlined by the reviewer, k=3 (Easy) and k=5 (Hard). In short, perfect retrieval always returns $k$ documents, prioritizing first the document of interest (if present), then documents from the same FARAD group, and finally all other documents. We note that this can easily result in documents from different sources or FARAD groups being present in the context. We hope this helps clear up the misunderstandings.
>
> **Q2: Do baselines perform better in the Hard setting with imperfect retrieval?**\
> This is an interesting question and a reasonable hypothesis. Prompted by this question, we ran a new experiment, evaluating all 3 baselines with end-to-end RAG retrieval (as in Sec. 5.3 and App. A.3), Llama3.1-70B, FARAD-Hard setting, and Def-P system prompt, on 5 seeds. Our results are very similar to the corresponding section of Fig. 4 which uses perfect retrieval, i.e. all baselines fall short, FACTS by having 100% FPR, and AAG and SIB by having 0% TPR. This demonstrates that our main experiments are a valid proxy for imperfect RAG retrieval, matching our prior result (L453) that shows 93.6% agreement between the two retrievers. We will extend this new experiment and include a more comprehensive version of the results in the next iteration of the paper.
>
> **Q3: Does the FARAD generation pipeline reflect real-world data well?**\
> While we certainly agree that this can be further improved, we believe FARAD is a solid baseline for future work in terms of how well it reflects realistic cases, greatly improving over the prior state where there were no suitable datasets that (1) are provably not leaked and (2) contain fact redundancy. In a newly added App. F (referenced from Sec 3.1), we recap the key design decisions of FARAD, and explicitly point out several promising directions in which FARAD can be improved in future work, including the approach suggested by the reviewer. We hope this helps better position our dataset contribution and makes building on top of our work easier.
>
> **Q4: Is Def-P applied by the RAG provider or the model owners via queries? If the first interpretation is correct, why would a RAG provider use Def-P to instruct the model not to answer questions?**\
> The first interpretation is indeed correct: Def-P is applied from the RAG providers’ and not the data owners’ side. However, Def-P does not instruct the model to never answer questions related to the retrieved content—this seems to be a misconception, likely caused by our wording on L323. In the new revision, we have added a pointer to the full Def-P prompt, and clarified that it actually instructs against verbatim text regurgitation and complying with user requests to learn about the exact contents of the LLM context. Both of these only prevent unintended/undesirable uses, with the latter one being analogous to the ongoing attempts to protect proprietary system prompts.

---

### Author Response · Authors · 2024-11-21
**General Response**

We thank the reviewers for their feedback and the positive evaluation of our work. We are pleased to hear that they unanimously consider our contributions important and meaningful, filling a research gap and establishing a strong foundation for future work (Lfcg, WHEe, nePY, naE4). We are glad that they also appreciate the novelty of our work and the thoroughness of our experimental evaluation (Lfcg, WHEe, naE4).

We have uploaded an updated version of the manuscript (new content marked in purple), and responded to all questions and concerns raised by the reviewers in individual responses below. We are happy to further engage in the discussion in case of follow-up questions.

---

### Author Response · Authors · 2024-11-25

We thank the reviewers again for their initial feedback. As we have not yet received replies to our rebuttal and the discussion period is closing shortly, we would like to kindly ask the reviewers to let us know if our responses addressed their concerns, and raise any follow-up questions.

---

### Meta-Review · Area_Chair_uyhw · 2024-12-20

**Metareview:**

The paper studies the problem of unauthorized use of datastores in RAG frameworks. The paper makes the following contributions: (a) a dataset FARAD for the RAG-DI problem, (b) a method WARD that uses llm watermarking to identify dataset usage. This is a valuable contribution that formalizes a novel problem in this space, and provides a good benchmark for future work to build on.

**Additional Comments On Reviewer Discussion:**

Reviewers (WHEe, Lfcg) raise the issue of whether the proposed setting actually emulates realistic scnearios. the authors address these questions sufficiently in my opinion.

---

### Decision · Program_Chairs · 2025-01-22

Accept (Poster)